# Optimized single-step optical clearing solution for 3D volume imaging of biological structures

Kitae Kim[1], Myeongsu Na[1], Kyoungjoon Oh[1], Eunji Cho[1], Seung Seok Han [2] & Sunghoe Chang [1,3 ✉]

Various optical clearing approaches have been introduced to meet the growing demand for 3D volume imaging of biological structures. Each has its own strengths but still suffers from low transparency, long incubation time, processing complexity, tissue deformation, or fluorescence quenching, and a single solution that best satisfies all aspects has yet been developed. Here, we develop OptiMuS, an optimized single-step solution that overcomes the shortcomings of the existing aqueous-based clearing methods and that provides the best performance in terms of transparency, clearing rate, and size retention. OptiMuS achieves rapid and high transparency of brain tissues and other intact organs while preserving the size and fluorescent signal of the tissues. Moreover, OptiMuS is compatible with the use of lipophilic dyes, revealing DiI-labeled vascular structures of the whole brain, kidney, spleen, and intestine, and is also applied to 3D quantitative and comparative analysis of DiI-labeled vascular structures of glomeruli turfs in normal and diseased kidneys. Together, OptiMuS provides a single-step solution for simple, fast, and versatile optical clearing method to obtain high tissue transparency with minimum structural changes and is widely applicable for 3D imaging of various whole biological structures.

[1] Department of Physiology and Biomedical Sciences, Seoul National University College of Medicine, Seoul 03080, South Korea. [2] Department of Internal Medicine, Seoul National University Hospital, Seoul 03080, South Korea. [3] Neuroscience Research Institute, Seoul National University College of Medicine, Seoul 03080, South Korea. ✉email: sunghoe@snu.ac.kr

There is a growing demand in the biological field for 3D volume imaging deep in biological tissues, but the optical heterogeneity of the tissue components results in opacity, which causes scattering of light as it penetrates the tissue, and makes 3D deep tissue imaging highly challenging[1,2]. A simple way to bypass tissue opacity is to section the thick tissue into several thin slices. Conventional tissue sectioning, however, is a laborious and time-intensive procedure and often causes tissue deformation during section preparation.

Recent surges of various 3D tissue clearing techniques open a new era in section-less 3D deep tissue imaging[3]. These methods can be classified into organic solvent-based clearing methods or aqueous-based clearing methods. Organic solvent-based methods including Benzoic Acid Benzyl Benzoate[4,5], dibenzyl ether[6], 3D imaging of solvent-cleared organs [7], ultimate DISCO[8] use organic reagents of high refractive index (RI), to achieve rapid and high transparency but, there are serious disadvantages in that the tissues are significantly shrunk and most of the fluorescence signals of the proteins disappear.

Aqueous-based methods generally show good fluorescence preservation and maintain tissue size within a reasonable range. They utilize (1) simple immersion in a solution containing high RI materials such as $-2,2'$-thiodiethanol[9,10], (2) hyperhydration of the sample by urea and lipid removal (3) hydrogel embedding followed by active or passive removal of lipid. Immersion in high RI solutions such as fructose-based SeeDeepBrain (SeeDB)[11] is effective for passive clearing of relatively thin samples, but due to high viscosity, its application to thick samples is limited. To avoid these issues, low viscosity alternative-based methods such as diatrizoic acid (FocusClear)[12,13], or iohexol (RIMS)[14] have been reported. However, they also suffer from long clearing time or insufficient clearing efficiency due to poor tissue penetration. For better clearing performance, urea-based hyperhydration followed by detergent-mediated lipid removal has been adopted in Sca*le*[15] and Sca*l*eS[16] and clear, unobstructed brain/body imaging cocktails and computational analysis (CUBIC)[17,18]. FRUIT[19] combines SeeDB and urea to improve tissue penetration and clearing. These methods improved clearing efficiency to some extent but still suffer from slow clearing, loss of protein content, and complexity of handling. Clear lipid-exchanged acrylamide-hybridized rigid imaging/immunostaining/in situ hybridization-compatible tissue hydrogel (CLARITY)[20,21] and its variants[10,22] use ionic detergents such as sodium dodecyl sulfate (SDS) and hydrogel embedding to clear tissues. Although it could achieve high tissue transparency, it requires specialized devices for electrophoretic lipid removals. Another drawback of the lipid removal methods is that they are not compatible with the use of lipophilic dyes such as DiI, which has been widely used to trace neuronal circuitries between brain regions and visualize vascular structures in whole organs. Recently m-xylylenediamine (MXDA)-based Aqueous Clearing System (MACS)[23] was introduced to address these issues and is also compatible with DiI but requires sequential incubation in three different compositions of MXDA and sorbitol. Besides, it shows only moderate levels of transparency in the visible wavelength range compared to other previous clearing methods.

Evidently, each clearing technique has its own specific goals but also suffers from various shortcomings, including low/moderate transparency, long incubation time, the complexity of processing, tissue deformation, or fluorescence quenching. Nevertheless, no single solution has yet been developed that overcomes the shortcomings of previous clearing methods and best meets all of the above aspects. Here, we developed OptiMuS (Optimized single-step optical clearing Method that preserves fluorescence and Size of the sample for 3D volume imaging) that is newly formulated by combining optimized concentrations of urea, iohexol, and sorbitol. We found that OptiMuS overcomes the shortcomings of the existing aqueous-based clearing methods and achieves high transparency of thick tissues and whole organs rapidly while preserving the size and endogenous fluorescent signals. OptiMuS enabled 3D volumetric imaging of neural structures of *Thy1*- EYFP transgenic mice as well as immunostained rodent brains. Moreover, OptiMuS is fully compatible with the use of lipophilic dyes, thus revealing DiI-labeled vascular structures of the whole brain, kidney, spleen, and intestine. We further showed that by combining a software-assisted automated 3D morphological analysis by DXplorer, OptiMuS achieves 3D quantitative and comparative analysis of DiI-labeled vascular structures of glomeruli turfs in normal and diseased kidneys. Together, OptiMuS provides a single-step solution for a simple, fast, and versatile optical clearing method that could be widely applicable for 3D volume imaging of various biological structures.

## Result

**OptiMuS enables rapid clearing of various tissues without sample deformation.** To develop the optimized clearing solution and overcome the shortcomings of the existing aqueous-based clearing methods to achieve high transparency while preserving the size of the sample, we combined iohexol with urea and D-sorbitol. Iohexol has a high RI (RI = 1.46) but low viscosity thus an iohexol-based clearing has frequently been applied to improve the transparency of the thin tissues[24]. However, its application to thick tissues was limited due to its low penetration capability[25]. We hypothesized that combining iohexol with urea's hyperhydration capability to reduce light scattering and improve tissue penetration, and D-sorbitol's gentle clearing and sample preservation capability could facilitate the clearing performance in thick tissues.

We first optimized the concentration of urea together with D-sorbitol since a high concentration of urea facilitates tissue clearing and penetration but over-hyperhydration by a high concentration of urea causes significant tissue deformation[11]. We found that 4 M urea with 10% D-sorbitol rendered considerable tissue transparency while maintaining the size of tissues (~58% transparency, Supplementary Fig. 1a–c). When we combined them with 75% iohexol, we found that it achieved higher tissue transparency (~75% transparency) without causing a change in size (Supplementary Fig. 1d, e). This newly formulated solution has a RI of 1.47 (Supplementary Table 1) and we named it OptiMuS.

We found that OptiMuS outperformed most previous aqueous-based clearing methods when considering tissue transparency, clearing speed, and size preservation together. OptiMuS rendered 1-mm thick rat brain tissues highly transparent within 1.5 hr while maintaining the size of samples with negligible change (0.93 ± 1.1% shrinkage). CUBIC, Sca*l*eS, Sca*l*eSQ(0), MACS, and FOCM[26] caused either significant expansion or shrinkage of the samples (Fig. 1a–c). In addition, OptiMuS achieved better transparency than other methods at wavelengths from 400 to 800 nm (Fig. 1d). We quantitatively evaluated the clearing capability of OptiMuS by comparing size change and transparency with other clearing methods and found that OptiMuS is the best of the clearing methods and renders high transparency and size preservation (Fig. 1e). Using high-resolution confocal imaging, we compared the tortuosity of the dendrite segments before and after OptiMuS clearing, and measured the structural similarity index measure (SSIM) of the soma, and confirmed that OptiMuS did not cause the deformation to the ultrastructure of dendrites and soma (Supplementary Fig. 2).

We also tested the performance of OptiMuS in a thinner rat brain section (200-μm thick) or whole mouse brain and confirmed that OptiMuS outperformed other aqueous-based

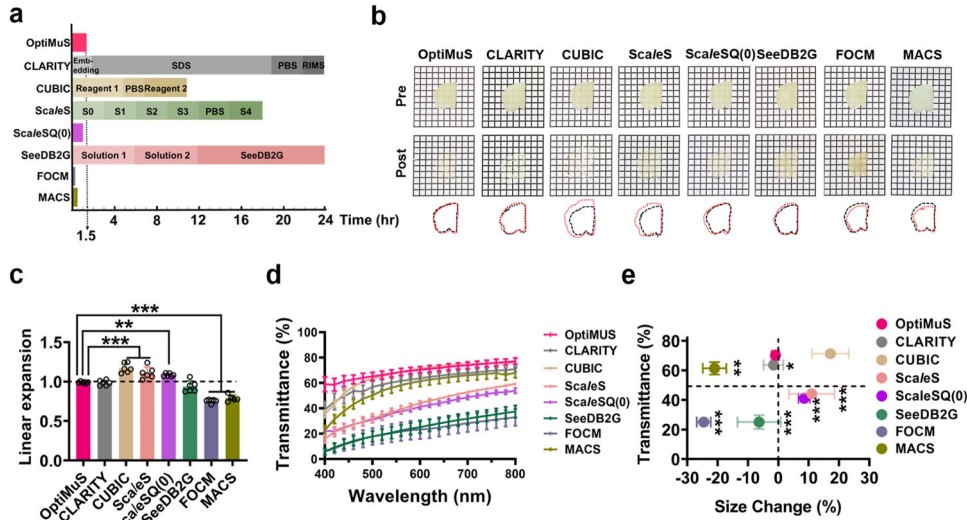

**Fig. 1 OptiMuS outperforms other optical clearing methods when considering transparency, clearing rate, and size preservation together. a** The timelines of OptiMuS, CLARITY, CUBIC, ScaleS, ScaleSQ(0), SeeDB2G, FOCM, and MACS for clearing of 1-mm thick rat brain sample. **b** Bright-field images of pre-and post-cleared by each method in **a**. Overlapped images of outlined pre- and post-cleared brain tissues (black: pre-, red: post-cleared). Grid size = 1.5 × 1.5 mm. **c** Quantitative comparison of the linear expansion after processing with each clearing method (n = 6). **d** Transmittance scan curves at 400–800 nm after clearing 1-mm thick rat brain samples by each method (n = 4). **e** Transmittance at 600 nm vs. size change graph for each method. The data were shown as the mean ± SD. ***$p < 0.001$, **$p < 0.01$, *$p < 0.05$.

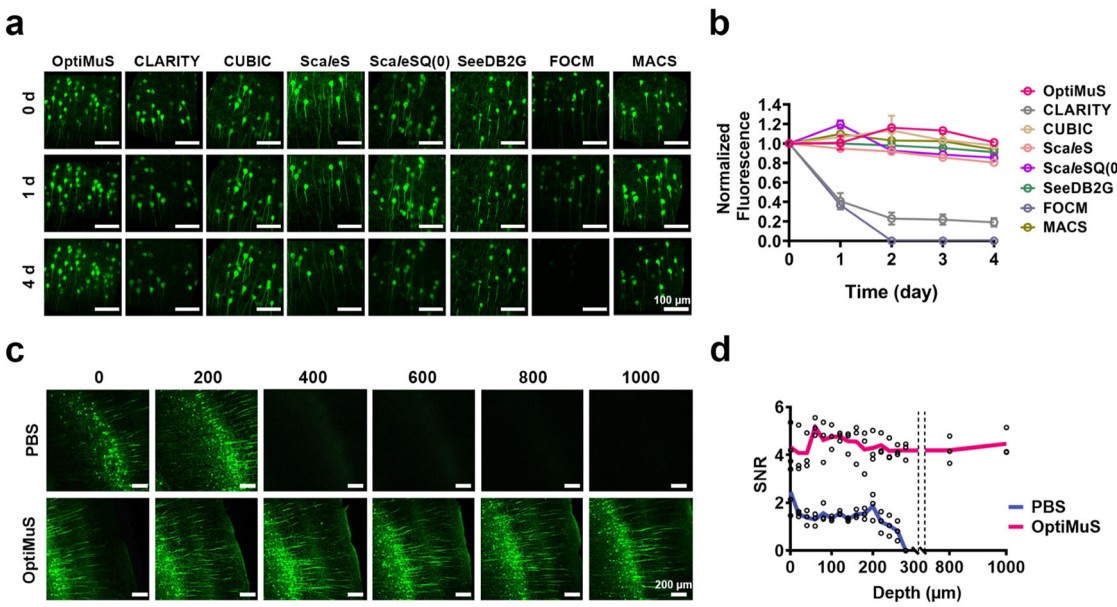

**Fig. 2 OptiMuS preserves the fluorescence signal of endogenous proteins. a** Fluorescence images of the endogenous EYFP signals of 50-μm thick *Thy1*-EYFP brain slices were taken daily at the same position after clearing with each method. Scale bar = 100 μm. **b** Normalized fluorescence intensity curves of images taken daily after each clearing method, normalized to the signals on day 0. The data were shown as the mean ± SD (n = 3). ***$p < 0.001$, **$p < 0.01$, *$p < 0.05$. **c** Fluorescence images of 1-mm thick *Thy1*-EYFP brain slices at various depths after clearing with OptiMuS and PBS for 1.5 h. Scale bar = 200 μm. **d** SNR values over the depth of imaging in **c** (n = 3).

clearing methods when considering both tissue transparency and size preservation (Supplementary Figs. 3 and 4). Besides brain tissues, OptiMuS also effectively rendered high transparency of various mouse and rat organs including the liver, lung, and intestine as well as heme-rich tissues such as the heart, kidney, and spleen (Supplementary Fig. 5).

**OptiMuS preserves fluorescence signals and improves imaging quality.** We then tested whether OptiMuS could preserve the endogenous signals of fluorescent proteins. The brain samples from

*Thy1*-EYFP transgenic mice were processed with various clearing methods and images were taken daily at the same position over time (Fig. 2a). The results showed that more than 90% of the fluorescence signal was preserved even after 4 days with OptiMuS, which was significantly better preservation compared to CLARITY and FOCM while comparable to the rest (Fig. 2b). FOCM is known to preserve 86% of the EYFP fluorescence up to 11 days[26], but was failed to preserve the fluorescent signal after 1 day in our hands.

We next measured the image quality of OptiMuS-cleared samples. After clearing of 1 mm-thick brain tissues from *Thy1*-

EYFP transgenic mice, we found that fine details of neuronal processes were clearly observed throughout the entire depth of imaging without compromising image quality (Fig. 2c). We quantitatively measured the signal-to-noise ratio (SNR) over the depth of imaging after clearing. In the PBS-treated sample, the SNR value decreased and the fluorescence signal became invisible over a depth of 280 μm. On the other hand, after the OptiMuS clearing, the SNR value started at a higher value (4.31) and did not decline over the entire depth of imaging up to 1 mm (Fig. 2d).

**OptiMuS enables 3D visualization of cellular structures in a single-cell resolution.** Next, OptiMuS was applied to clear whole brains from *Thy1*-EYFP transgenic mice. The whole mouse brains were cleared by OptiMuS for 3 days, and 3D volume imaging was carried out with light-sheet fluorescence microscopy (LSFM) (Fig. 3a–d). The fine neural structures including cell bodies and neurites were well observed at various depths in the brain, including the cortex and striatum (Fig. 3e, f).

OptiMuS also was applicable for clearing of the intestine and brain samples of *ChAT*-Cre-tdTomato transgenic mice (Fig. 3g, i). Cholinergic nerves were well resolved throughout the intestine, revealing myenteric plexus and submucosal plexus around the intestinal walls (Fig. 3h). Cholinergic neurons in the nucleus accumbens of the brain were also observed in a single cell resolution over the entire depths (Fig. 3j).

OptiMuS can also be used as a RI matching solution for high-resolution 3D volume imaging of pre-stained samples. Mouse brain and intestine that were cleared by CLARITY, stained with lectin, or anti-GFAP and anti-Tuj-1 antibodies. The stained samples were then immersed in OptiMuS, then 3D volume images were obtained. Astrocytes and blood vessels in brain samples stained with anti-GFAP antibody and lectin as well as enteric nerve processes stained with anti-Tuj-1 antibody were well observed throughout the entire depth of imaging (Supplementary Fig. 6).

**OptiMuS preserves the DiI signal in the vasculatures after clearing.** Transcardial perfusion of lipophilic DiI solution is an effective method to label vasculatures in various whole organs especially the spleen and kidney[27]. However, many clearing techniques are generally not compatible with the use of lipophilic dyes due to their lipid removal properties. Since OptiMuS is an aqueous-based clearing method based on RI matching without delipidation, we tested whether OptiMuS could preserve DiI signals after clearing. Mice were transcardially perfused with DiI to label vasculatures of various organs including the brain, intestine, and spleen, and after sacrifice, these organs were isolated and cleared by OptiMuS. We found that after clearing with OptiMuS, DiI-labeled vascular structures in the whole mouse brain were well resolved with LSFM imaging (Fig. 4a, b). We then halved the brain axially or sectioned coronally, and imaged with LSFM to visualize internal vascular structures in the transverse section, and confirmed that OptiMuS cleared the whole brain thoroughly while preserving DiI signals in the vascular structures (Fig. 4c–h). High-resolution confocal imaging of a 2800-μm thick sample from the DiI-labeled brain showed that fine details of the vascular tree structure were well resolved over the entire depth of imaging (Supplementary Fig. 7). We also obtained 3D images of the vascular structures of the intestine and spleen which were transcardially labeled with DiI. The detailed vascular tree networks throughout the intestine and whole spleen were clearly visualized, and the 3D depth-coded images showed that the distribution of blood vessels along the z-position can be readily traced in three-dimension (Fig. 4i–l).

**OptiMuS enables quantitative 3D analysis of the kidney vascular system.** Murine nephrotoxic nephritis (NTN) is an acute model of human glomerulonephritis and chronic kidney disease[28,29]. The previous histological study showed that the NTN induced glomerular mesangial expansion over time[28] although no detailed 3D structural analysis was performed. We sequentially performed transcardial DiI perfusion, OptiMuS clearing, and LSFM imaging to display the glomeruli and vascular structures in normal whole kidneys and 2-week-old NTN model mouse kidneys (Fig. 5a). OptiMuS clearly visualized fine details of DiI-labeled glomerular structures in the whole kidney (Fig. 5b, c).

To accurately analyze the alteration in glomeruli morphology in 3D, we used DXplorer, a 3D morphology analysis program that we recently developed[30]. DXplorer was originally developed to analyze and categorize dendritic spine morphology and due to the structural similarities between the dendritic spine of neurons and the glomeruli of the kidney (with a round-shape head and a connecting neck), we expected that it can be applied to analyze the glomerulus structure of the kidney. Besides, algorithms for feature rendering and detection of DXplorer are not limited to a specific structure but can be applied to any structure that protrudes from the base structure such as a head and connecting neck[30].

The DiI-labeled whole mouse kidneys from normal and NTN model were imaged as z-stacks at 4 μm intervals using an LSFM and the obtained images were transferred to the DXplorer software for 3D mesh construction and 3D feature extraction (Fig. 5d, e). We found that compared to normal kidneys, NTN caused a significant increase in glomerulus volume, maximum head (hMax), and neck diameter (nMax) (Fig. 5f). To check the accuracy of DXplorer's automatic detection and feature extraction, we manually isolated individual glomeruli and constructed meshed images using Meshlab, an open-source system for processing and editing 3D triangular meshes[31] and then measured the volume and the length of glomeruli. We compared these values with DXplorer's and found that there is no significant difference between them (Supplementary Fig 8). This result indicates that OptiMuS combined with DXplorer can compare and objectively analyze detailed 3D structural features of kidneys in normal and diseases states.

## Discussion
We developed OptiMuS, a simple and rapid single-step optical clearing method based on iohexol, urea, and D-sorbitol. We showed that OptiMuS outperformed most previous optical clearing methods when considering tissue transparency, clearing rate, size maintenance, and endogenous fluorescence preservation together. OptiMuS enabled 3D volumetric imaging of neural structures of *Thy1*-EYFP transgenic mice as well as immunostained rodent brains and was fully compatible with various labeling probes including lipophilic dyes (Supplementary Table 2).

Tissue clearing technology aims to reduce light scattering in order to image large-volume tissues. Each method has its own specific goals and pros and cons. Solvent-based dehydration processes and aqueous-based hyperhydration processes improve transparency, but these processes still require long incubation to achieve sufficient transparency and often cause shrinkage or expansion of the sample[32]. For example, Sca*l*eSQ(0) that was previously used for brain slice clearing[16] induces significant changes in size due to the high concentration of urea (9.1 M), which was also observed in Fig. 1c. MACS uses MXDA as a substitute for urea and is known to clear large volumes of tissues within a relatively fast time[23]. However, it becomes yellowish during clearing and requires serial incubation in solutions

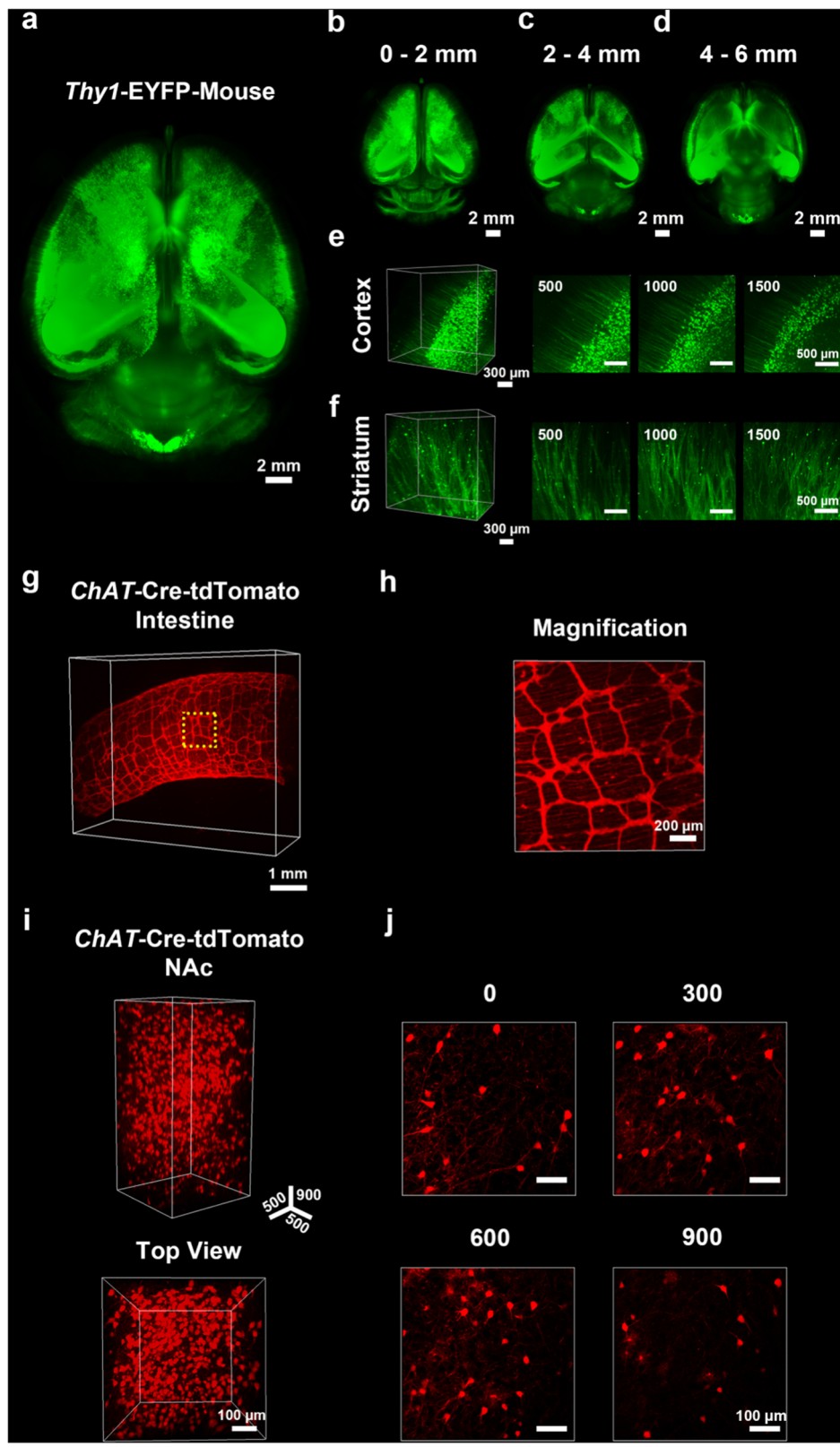

consisting of three different compositions. FOCM is a DMSO (dimethyl sulfoxide) based optical clearing method that rapidly clears thin tissues and is known to preserve endogenous fluorescence for up to 11 days[26]. We, however, found that it induces shrinkage of the sample and lost most of the fluorescence after 1-day incubation (Figs. 1, 2 and Supplementary Figs. 3, 4). We followed exactly the published protocol and repeated it several times with no success. The cause of this discrepancy is currently unknown. Regarding the size change, the difference between FOCM and ours is that FOCM used mouse brain slices (C57BL/6, 9 weeks old) while we used rat brain slices (Sprague-Dawley, 3 weeks old). In addition, the FOCM reported the deformation value of the brain slice obtained from one hemisphere, and which depth of the coronal plane was used was not explicitly

**Fig. 3 OptiMuS enables 3D visualization of neural structure networks. a** Top view of maximum intensity z-projection image of the whole brain from *Thy1*-EYFP transgenic mouse cleared by OptiMuS for 3 days. Scale bar = 2 mm. **b–d** Maximum intensity z-projection images of the optical sections between 0 and 2 mm (**b**), 2–4 mm (**c**), and 4–6 mm (**d**). Scale bar = 2 mm. Parts of magnified 3D projection images and optical section images at 500, 1000, and 1500 μm of the cortex (**e**) and striatum (**f**) in **a**, respectively. Scale bar = 300 μm, 500 μm respectively. **g** 3D reconstruction image of the intestine from *ChAT*-Cre-tdTomato transgenic mouse after OptiMuS clearing for 10 hr. Scale bar = 1 mm. **h** Magnified maximum z-projection image of the area enclosed by a rectangle in **g**. The networks of cholinergic neurons including the myenteric plexus and submucosal plexus were clearly visualized. Scale bar = 200 μm. **i** 3D reconstruction images of the nucleus accumbens (NAc) region of *ChAT*-Cre-tdTomato mouse brain after OptiMuS clearing for 1 h. The 3D axis is expressed in μm. **j** Optical section images at each depth. Scale bar = 100 μm.

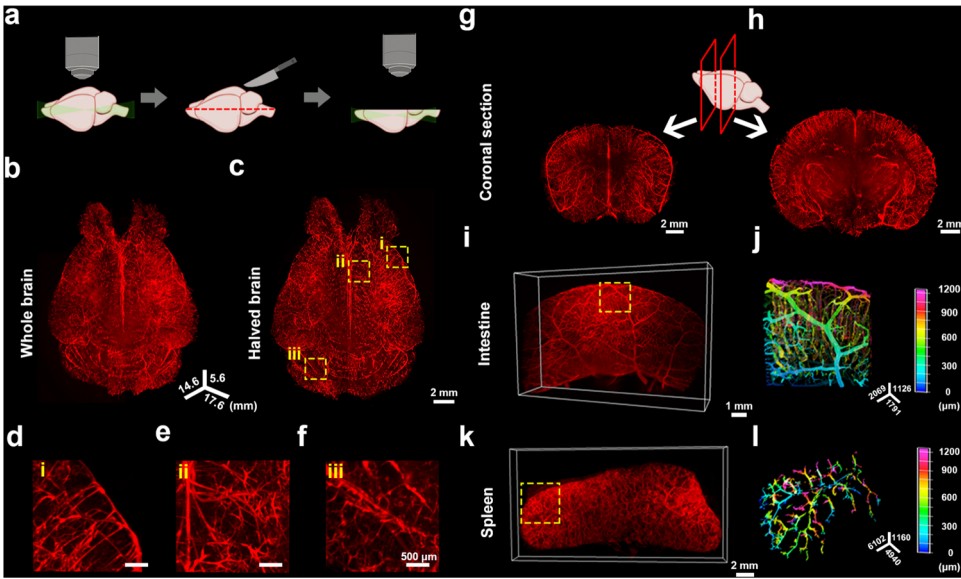

**Fig. 4 OptiMuS visualizes DiI-labeled 3D vascular structures of various whole organs. a** The schematic diagram of DiI-labeled whole and axially sectioned brain imaging with LSFM after OptiMuS clearing for 3.5 days. **b** 3D reconstruction image of DiI-labeled whole mouse brain. **c** Top view of maximum intensity z-projection image of DiI-labeled mouse brain that was halved axially (See **a**). Scale bar = 2 mm. **d–f** Magnified images of the areas enclosed by rectangles in **c**. Scale bar = 500 μm. **g–h** The schematic diagram and 3D reconstruction image of DiI-labeled coronally sectioned brain imaging with LSFM after OptiMuS clearing for 3.5 days. **i** 3D reconstruction images of the vascular structures of DiI-labeled whole intestine (**i**) and spleen (**k**). Scale bar = 1 mm (**g**) and 2 mm (**h**). **j, l** Pseudo-color depth-coded images of the boxed regions in **g** and **h**, respectively.

described[26]. Since different anatomical brain regions are known to show somewhat differing clearing results, depending on the degree of myelination[33,34], we suspect the reason for this discrepancy might be due to the different regions used or the existence of the corpus callosum, which is a large bundle of myriad myelinated fibers that connect the two brain hemispheres.

We aimed to develop a single-step optical clearing method that is simple, rapid, and has a maximum transparency and minimum size distortion and developed a newly formulated optimized optical clearing method, OptiMuS. Compared to other clearing methods, OptiMuS has a slightly slower clearing rate than Sca-*le*SQ(0), FOCM, and MACS, but preserves the sample size much better than other methods (Fig. 1). Indeed, OptiMuS outperforms other aqueous-based clearing methods when considering both tissue transparency and size preservation. Besides, it is a simple, single solution method thus does not require any additional incubation in other solutions. One caveat with OptiMuS is that the degree of deformation is somewhat greater when clearing organs other than the brain (Supplementary Fig. 5). However, different organs have different compositions of lipids, proteins, and extracellular matrix, which mainly affect the performance of tissue clearing[35]. It means that the composition of the clearing solution and the timeline should be optimized for the best performance in each organ. Considering its excellent performance in brain tissue, we expect that OptiMuS performs well if its composition is organ-optimized according to the differences in each organ.

Lipophilic dyes including DiI are widely used to visualize cell membranes, neural micro-circuitry, and vasculature in various tissues[27]. However, most clearing methods that use detergent or organic solvent to remove lipids to increase clearing performance are incompatible with the use of lipophilic dyes. Since OptiMuS does not remove lipid, it successfully visualizes the vasculatures of various DiI-labeled organs. We further showed that by combining OptiMuS and DXplorer, the comparative and quantitative 3D morphological analysis of kidneys from normal and disease models is possible. Although we did not perform further analysis in this study, since the DXplorer can provide 10 different 3D features including volume, hMax, and hMin, one can obtain more insightful data regarding the 3D structures of interest[30] since its algorithm for feature rendering and detection are not limited to a specific structure but can be applied to any structure with a round-shape head and a connecting neck. More importantly, DXplorer can be trained to perform automatic machine-learning-based feature classification. Thus, using this feature, different shapes of diseased kidneys can be objectively classified, and these data can be used to monitor the progress of the diseases. For example, the kidney of the NTN disease model undergoes a series of progressions over time, such as mesangial expansion and fibrosis[28,29], which can be classified by DXplorer to objectively diagnose the disease state although it certainly needs further investigation and optimization.

Due to its simple procedure and excellent size preservation, OptiMuS would be applicable to staining-free optical imaging modalities such as optical coherence tomography (OCT) rather

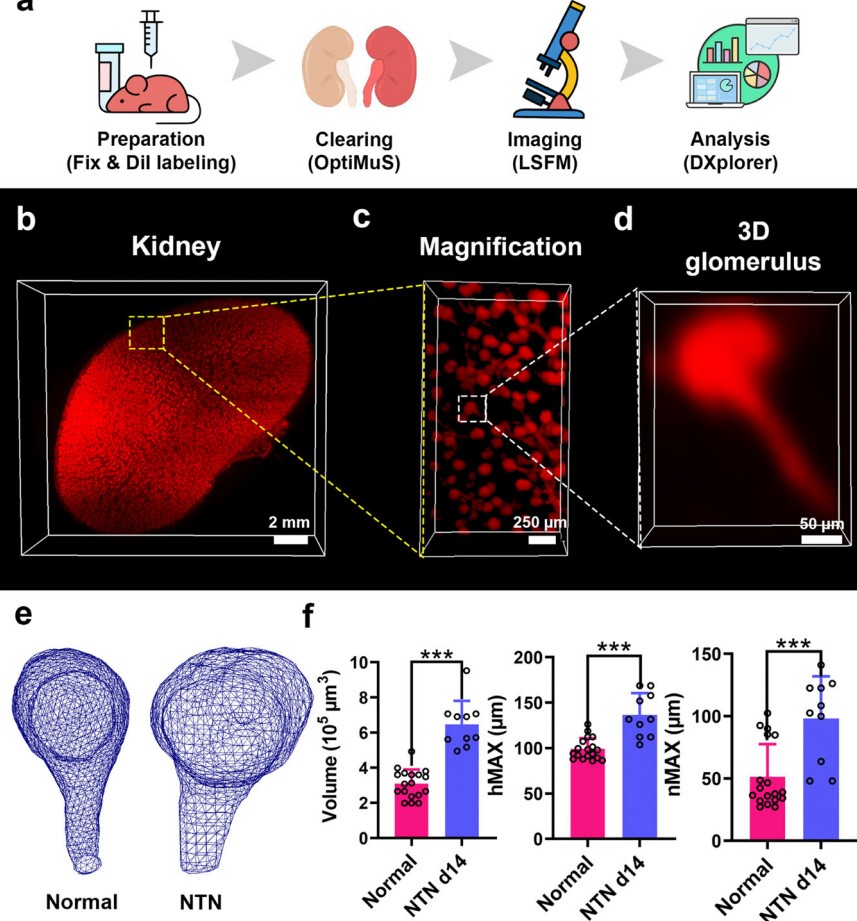

**Fig. 5 OptiMuS enables comparative 3D analysis of glomeruli structures of normal and diseased kidneys. a** The workflow for the comparative 3D analysis of DiI-labeled kidney glomeruli structures via OptiMuS and DXplorer. **b** 3D reconstruction image of DiI- labeled whole mouse kidney after OptiMuS clearing and LSFM imaging. Scale bar = 2 mm. **c** Magnified view of the boxed region in **b**. Scale bar = 250 μm. **d** 3D reconstruction image of a single kidney glomerulus in **c**. Scale bar = 50 μm. **e** Representative 3D mesh images of individual glomeruli of normal and NTN model. **f** 3D features including volume, hMAX, and nMAX were extracted from the 3D meshes of the normal glomeruli and NTN glomeruli and analyzed by DXplorer. hMax (maximum head diameter), and nMax (maximum neck diameter). $n = 18$ (normal) and 10 (NTN). The data are shown as the mean ± SD ***$p < 0.001$.

than optical microscopes based on the fluorescent contrast. Since OCT uses near-infrared light and endogenous contrast, it offers deep tissue morphology over the range of several hundred micrometers[36,37]. However, the imaging depth of OCT is still limited for visualizing diagnostic features over large areas or volumetric contexts. To overcome this limitation, the idea that uses tissue clearing reagents with OCT to reduce scattering and improve the OCT imaging depth has been experimentally investigated by several groups[38–40]. Despite the previous efforts, however, due to mal-preservation of tissue size and relatively slow clearing, tissue clearing-based OCT has only limited applications so far. Besides, too much tissue clearing would result in a decrease in reflectivity thus meticulous optimization is critical to balance these two factors. Although yet preliminary, we found that OptiMuS could further enhance OCT imaging depth and render label-free whole 3D organ image reconstruction. Moreover, the technical integration of OptiMuS and OCT would be well suited for the need for fast 3D histopathology[41], as it can exceptionally preserve the size of the tissue and can reduce scattering coefficient within a few minutes. Although further investigations are required to substantiate the validity, a fully engineered imaging protocol would certainly allow the investigation of 3D anatomical alterations associated with various diseases, and provide a paradigm shift towards digital histopathology.

In summary, the advantages of speedy clearing, nontoxicity, good fluorescence preservation, negligible size distortion, and simple protocol by OptiMuS can be widely applied to various organs. We expect OptiMuS to be optimized as the first choice of general optical clearing process to visualize 3D volume samples and to be used for 3D pathology analysis of various organs for diagnostic purposes.

## Method

**Animals**. Mice (C57BL/6N, 6–8 weeks old, male) and rats (Sprague-Dawley, 3weeks old, female) were used in the experiments. Brain sections from *Thy1*-EYFP H-line mice (6 months old, male) were provided by Dr. Chang Man Ha of the Korea Brain Research Institute. Brain and intestine samples from *ChAT*-Cre-tdTomato mice (8–12 weeks old, male) were provided by Dr. Hyung Jin Choi of the Seoul National University College of Medicine.

**Ethics statement**. Animal husbandry and all experimental procedures involving mice and rats were approved by the Institute of Animal Care and Use Committee (IACUC) guidelines of Seoul National University (SNU-200904-2-4). Transgenic animal treatment followed animal care guidelines approved by the IACUC of the KBRI (IACUC-18-00018).

**Sample preparation**. Adult mice and rats were deeply anesthetized with a mixture of zoletil (30 mg kg$^{-1}$) and rompun (xylazine; 10 mg kg$^{-1}$). The animals were transcardially perfused with 100 ml of cold PBS (phosphate-buffered saline), followed by 50 ml of 4% (w/w) PFA (paraformaldehyde; Sigma, St.Louis, MO) using a peristaltic pump (MP-100, Tokyo Rikakikai) at the speed of 5 ml/min. All

harvested samples were immersed overnight in 4% PFA at 4 °C on a rotator for post-fixation and then washed with 50 ml PBS in a 4 °C rotator at 20-min intervals for 1 h to remove residual PFA before clearing. For the experiments, all samples were cut into sections using a vibratome (Leica VT1200s, Germany).

**Preparation of OptiMuS solution**. To perform the clearing of mouse organ tissues, 100 mM Tris (Tris-(hydroxymethyl)-aminomethane; Sigma) and 0.34 mM EDTA (Ethylenediaminetetraacetic acid; Junsei, Japan) were dissolved in ddH₂O and titrated to pH 7.5 to prepare Tris-EDTA solution. Histodenz solution (a.k.a. Iohexol, Sigma) was prepared by dissolving 75% (w/v) histodenz in Tris-EDTA solution at 60 °C until totally transparent. After complete dissolution, 10% (w/v) D-sorbitol (Sigma) and 4 M urea (Thermo Scientific, Waltham, Massachusetts) were subsequently dissolved in the mixture at 60 °C. The solution was then cooled to RT (room temperature). All clearing procedures were performed at 37 °C and the solution was stored at 4 °C for later use.

**Quantitative measurement of the linear expansion**. For the measurement of the sample size change, bright-field images were obtained with a commercial camera (NEX-3, Sony, Japan). The contours of three brain samples were extracted using the 'outlines' tool of the ImageJ/Fiji program (National Institute of Health, USA) after thresholding images and the outlines of pre- and post-cleared images were manually aligned. The linear expansion value was calculated by the square root of size change in the area. The size change according to urea concentration (Supplementary Fig. 1) and when various organs were cleared (Supplementary Fig. 3) was determined by the ratio between areas of pre- and post-cleared images.

**Quantitative measurement of transmittance**. For the measurement of transmittance, the transparency of 1-mm thick rat brain slices and whole mouse brains were measured using a spectrophotometer (UV mini-1240, Shimadzu, Japan). Briefly, the cleared brain was placed in a cuvette and positioned so that light would pass through the cortical region, and the transparency of the slices was measured from 400 nm to 800 nm. The blank value was measured in the presence of a clearing reagent without a sample and the transmittance of the sample was normalized to a blank value. To measure the transmittance according to urea, iohexol concentrations of 1-mm thick rat brain slices and 200-µm thick rat brain slices, bright-field images of rat brain sections were color-inverted using the reverse function of the Fiji program. Then, the intensities of the grid outside the sample and the inner grid visible through the sample were obtained. The intensity value of the inner grid was normalized to that of the external grid to measure relative transmittance. The transparency expressed the ratio of the inner grid to the outer grid as a percentage.

**Quantification of normalized fluorescence intensity**. For the analysis of EYFP fluorescence intensity retention, *Thy1*-EYFP transgenic mouse brains were sliced into the 50-µm thickness and EYFP fluorescence images were taken daily up to 4 days at the same position after clearing with each method. After that, maximum intensity projection of all images was performed, and the mean intensity of neuronal cell bodies ($n = 8-10$) and the surrounding background intensity were measured manually using the freehand selection tool in Fiji. All fluorescence signals were normalized based on day 0 after subtracting the background value.

**Measurement of SNR**. To calculate the SNR, *Thy1*-EYFP transgenic mouse brains were sliced into the 1-mm thickness and EYFP fluorescence images were taken at 5 µm depth intervals before and after clearing using a confocal microscope with a Plan Apochromat 10× objective lens (NA 0.5, WD 5.5 mm). The laser power was maintained at 150 µW regardless of depth. Then, the acquired images were transferred to Fiji for further analysis. From the optical sectioned single image at each depth, we selected five dendrites randomly, drew a line perpendicular to the dendrite axis, and measured the fluorescence intensities of the dendrites and adjacent surrounding areas. From the surface to a depth of 280 µm, we measured the SNR value at 20 µm intervals, and then the SNR values at depths of 600, 800, and 1000 µm were measured and plotted.

**CLARITY (SDS-based) clearing**. The CLARITY solution was prepared by mixing 4% SDS, 50 mM lithium hydroxide (LiOH) (39225-0450, Junsei Chemical, Tokyo, Japan), and 25 mM boric acid (15663, Sigma) in ddH₂O at pH 9.0. The CLARITY clearing was carried out according to the previously published protocol. The cleared samples were washed with 50 ml PBS twice a day at 37 °C.

**Immunostaining**. Brains (700-µm thick block) and intestines (whole organ) were cleared with CLARITY protocol and then immunostained. Antibodies were used in this study - Primary antibody: Anti-GFAP (ab53554, Abcam, Cambridge, UK), Anti-beta III Tubulin antibody (ab18207, Abcam); Secondary antibody: Alexa Fluor 488 Donkey anti-Goat IgG (H + L) (A11055, Invitrogen, Carlsbad, CA), Alexa Fluor 488, F(ab') 2-Goat anti-Rabbit IgG (H + L)(A11070, Invitrogen); Dyes: DyLight 594 labeled Lycopersicon Esculentum (Tomato) Lectin (DL-1177, Vector Labs, Burlingame, CA).

For immunostaining, the fixed sample was permeabilized in a pretreat solution, the mixture of 20% (w/v) DMSO (Sigma) and 2% (w/v) Triton X-100 (Sigma), in PBS at 37 °C for 4–5 h. After that, the sample was washed in PBS, transferred to a blocking solution, the mixture of 10% (w/v) BSA (bovine serum albumin; Bovogen, Australia) in a pretreat solution, and incubated at 37 °C for 4–5 h. After immunostaining with primary antibody (1:250 dilution) in PBS by rotating at 37 °C for 2–3 days, the stained samples were washed with PBS at RT for 1–2 h, then immunostained with secondary antibody (1:500 dilution) in PBS and incubated at 37 °C for 2–3 days. Stained samples were stored in PBS at 4 °C before clearing.

**DiI labeling via transcardial perfusion**. To prepare DiI stock solution, 150 mg DiI powder (1,1′-Dioctadecyl-3,3,3′,3′-tetramethylindocarbocyanine perchlorate, ThermoFisher Scientific) was dissolved in 50 mL of 100% ethanol (Sigma) and stored in a −20 °C light blocked environment. DiI working solution was made by mixing 1 mL of DiI stock solution with 40 mL of 5% (w/v) glucose (Sigma) and 10 mL of 0.01 M PBS. The working solution should be prepared without light exposure. The transcardial perfusion was performed by inserting a needle into the left ventricle and snipping the right atrium and perfused with 50 mL of cold PBS, followed by 40 mL of DiI working solution at a flow rate of 2 mL/min. During DiI perfusion, the ears, nose, and feet of the mouse turn slightly red. Lastly, the mice were perfused with 50 ml of 4% PFA and the organs were passively post-fixed overnight at 4 °C.

**NTN induction**. Mice (8-weeks old; $n = 3$) were pre-immunized by intraperitoneal injection with 200 µl mixture of 18 µl sheep gamma globulin (Jackson ImmunoResearch), 100 µl complete Freund's adjuvant (Merck, Germany), and 82 µl PBS. After 4 days, mice were intravenously injected with 200 µl NTS (anti-GBM serum; PTX-001 GBM serum, Probetex, Inc.) in a 1:1 mix with PBS. Mice were sacrificed under anesthesia 14 days after NTS administration to collect kidney samples.

**Glomerulus classification and analysis**. The DiI-labeled whole mouse kidneys (normal $n = 18$, NTN model $n = 12$) were imaged as z-stacks at 4 µm intervals using a light-sheet fluorescence microscope and the obtained images were transferred to the DXplorer software. Then, 3D images were interpolated along the z-direction to reduce the anisotropic resolution of the data. Then, the glomerulus was extracted using the geodesic active contour method applied to the binarized image using Otsu's thresholding method. Using an ellipse-fitting scheme, each glomerulus was separated and volume-rendered. The final triangular meshes were generated by the marching cube method for each detected glomerulus segment, and the surface was smoothed using a curvature flow smoothing algorithm. Because of the acquisition artifact and noise of the LSFM image data, however, incorrect or blurred glomeruli can often be detected especially around the edges of imaging planes. Therefore, using the intuitive and interactive proofreading feature of DXplorer, we confined the analysis to the center of the kidney structure and also verified how well the glomeruli were detected by interacting with the 3D visualization, which finds a specific glomerulus through 3D interaction such as rotation, zooming, or panning. After verification, ten glomeruli per kidney were randomly selected and analyzed.

To measure 3D features, three anchor points, i.e., Central base point, Centroid, and Tip, were automatically extracted from the triangular mesh representing the glomeruli. The orientation of the glomerulus was then set as the glomerulus direction from the Central base point to Centroid. The vertices were divided into steps of a certain height along with the orientation of the glomerulus from the Central base point to Tip, and L was measured as the length of the curve connecting the center of the local vertices in each step. hMin and hMax were measured as the minimum and maximum diameters of the thickest part between Centroid and Tip. See ref. [32] for more details regarding DXplorer analysis. The GitHub link of DXplorer is available at https://github.com/hvcl/SpineAnalysis_public.

**MeshLab measurement of kidney glomerulus**. Tiff file format images taken with LSFM were converted to polygon file format(.ply) using ImageJ "3D viewer" plugin. The individual glomeruli were manually isolated using the '3D crop' function of Imaris and were imported to MeshLab, a 3D mesh-processing open-source software program. For accurate distance measurement, smoothing was performed using the Laplacian smoothing algorithm. The voxel size of the 3D mesh was determined by calculating the image size of the raw data using the "Scale, normalize" tool. Using the application of the tool "measures" in MeshLab software, the distance between the vertices at the farthest distance was manually measured. Based on the top view, the two points furthest from the center of the mesh (hMax, nMax) were manually selected and measured. To measure the volume of the mesh, the rendered mesh was volume-shaped and re-meshed using the "closed hole" tool to give it a volumetric shape. Then the volume of the glomerulus mesh was measured with the "geometric measure" tool.

**Other optical tissue clearing methods**. The clearing time for each thickness of the tissue was determined according to the corresponding original report.
CLARITY. To make a hydrogel monomer solution, the mixing reagent contains 4% PFA, 4% (w/w) acrylamide (Sigma), 0.05% (w/w) bisacrylamide (USB Corporation, Cleveland, OH), and 0.25% (w/w) VA-044 (2,2′-azobis [2-(2-imidazolin-

2-yl) propane] dihydrochloride; Wako Pure Chemical, Japan). For 1-mm thick rat brain clearing, the brain tissues were embedded at 37 °C in a vacuum for 2 h. After that, the sample was immersed in 4% (w/w) SDS (Bio-Rad, Berkeley, CA) solution, and passive CLARITY clearing was performed for about 17 h in a 37 °C incubator so that the sample becomes transparent. After sufficiently washing the sample in PBS for 3 h, the RI was adjusted by putting it in RIMS for 2 h at RT. For 200-μm thick rat brain clearing, samples were treated CLARITY buffer for 2 min at 37 °C. After being washed briefly, samples were immersed in RIMS for 2 min at RT.

**CUBIC.** Reagent 1—25% (w/w) urea, 25% (w/w) N,N,N′,N′-Tetrakis (2-hydroxypropyl) ethylenediamine (TCI, Japan) and 15% (w/w) Triton X-100 in ddH₂O. Reagent 2—50% (w/w) sucrose (JUNSEI), 25% (w/w) urea, and 10% (w/w) triethanolamine(Sigma) in ddH₂O. The clearing time for each thickness of the tissue was determined according to the original report[17]. For 200-μm thick rat brain clearing, Reagent 2 was treated for 2 min at 37 °C. For 1-mm thick rat brain clearing, Reagents 1 and 2 were sequentially treated. After incubating the samples in Reagent 1 for 5 h at 37 °C and washing for 2 h at RT, RI was adjusted with Reagent 2 for 4 h at 37 °C. For whole mouse brain clearing, Reagent 1 was treated for 7 days at 37 °C, and after washing for 1 day at 37 °C, RI was adjusted with Reagent 2 for 2 days at 37 °C.

**ScaleS.** ScaleS0—20% (w/v) sorbitol, 5% (w/v) glycerol (Sigma), 1% (w/v) N-acetyl-L-hydroxyproline (Sigma), 3% (v/v) DMSO, 1 mM methyl-β-cyclodextrin (Sigma) and γ-cyclodextrin (Wako Pure Chemmical) in PBS. ScaleS1 − 20% (w/v) sorbitol, 5% (w/v) glycerol, 0.2% (w/v) Triton X-100, 4 M urea in ddH₂O. ScaleS2—27% (w/v) sorbitol, 0.1% (w/v) Triton X-100, 8.3% (v/v) DMSO, and 2.7 M urea in ddH₂O. ScaleS3 − 36.4% (w/v) sorbitol, 9.1% (v/v) DMSO, and 2.7 M urea in ddH₂O. ScaleS4—40% (w/v) sorbitol, 10% (w/v) glycerol, 0.2% (w/v) Triton X-100, 20% (v/v) DMSO, and 4 M urea in ddH₂O. The clearing time for each thickness of the tissue was determined according to the original report[16]. For 1-mm thick rat brain clearing, samples were sequentially immersed in ScaleS0 to S4 solution. Samples were incubated in ScaleS0 to S3 for 3 h each, then washed for 3 h, and immersed in S4 for 3 h to adjust RI. All processes were conducted at 37 °C. For whole mouse clearing, samples were sequentially immersed in ScaleS0 to S3 for 1day each, then washed for 0.5 day, and immersed in S4 for 1.5 days at 37 °C to adjust RI.

**ScaleSQ(0).** ScaleSQ(0) solution—22.5% (w/v) sorbitol, 9.1 M urea in ddH₂O. For 200-μm thick rat brain slice clearing, ScaleSQ(0) solution was treated for 2 min at 37 °C. For 1-mm thick rat brain clearing, samples were immersed in ScaleSQ(0) for 1 h at 37 °C.

**FOCM.** The clearing time for each thickness of the tissue was determined according to the original report[26]. For 200-μm and 1-mm thick rat brain, FOCM solution consisting of 30% (w/v) urea and 20% (w/v) D-sorbitol in DMSO was used. For whole mouse brain clearing, FOCM solution consisting of 20% (w/v) urea and 30% (w/v) D-sorbitol in DMSO was used. After completely dissolution, 5% (w/v) glycerol was added and mixed thoroughly. Brain tissues were immersed in FOCM solution for 2 min (200-μm), 5 min (1-mm), and 2 days (whole brain) at RT to adjust RI, respectively.

**SeeDB2G.** Solution 1—1/3 Omnipaque 350 (Iohexol, GE Healthcare), 2% (w/v) saponin (Sigma), Solution 2 —1/2 Omnipaque 350, 2% (w/v) saponin, SeeDB2G - Omnipaque 350, 2% (w/v) saponin. Solution 1&2 and SeeDB2G solutions were used sequentially. The clearing time for each thickness of the tissue was determined according to the original report[25]. For 1-mm thick rat brain clearing, Solution 1&2 were treated for 6 h at RT, followed by SeeDB2G for 12 h at RT. For 200-μm thick rat brain clearing, SeeDB2G was treated for 2 min at RT.

**MACS.** MACS R0- 20%(v/v) MXDA (m-Xylylenediamine; TCI), 15% (w/v) D-sorbitol in ddH₂O MACS R1- 40% (v/v) MXDA, 30% (w/v) D-sorbitol in PBS, MACS R2- 40%(v/v) MXDA, 50% D-sorbitol in ddH₂O. For 200-μm thick rat brain clearing, samples were immersed in R1 solution for 2 min at RT. For 1-mm thick rat brain clearing, samples were sequentially treated with R0 for 20 min, and R1 and R2 for 10 min each at RT. For whole mouse brain clearing, samples were sequentially immersed in R0 for 1.5 days, and R1and R2 for 0.5 days each at RT.

**Imaging.**
1) Upright confocal microscope: The samples immunolabeled with dye or antibodies and SNR measurement samples were placed on a 35 mm petri dish, immobilized on a 22 mm glass coverslip, immersed in OptiMuS, and imaged with an upright confocal microscope (Nikon C2Si, Japan) with a Plan-Apochromat 10× lens (NA = 0.5, WD = 5.5 mm) using NIS-Elements software (Nikon).

2) Inverted confocal microscope: For the comparison of fluorescence signal preservation between clearing methods, the sample was set on a coverglass (HSU-0101152, MARIENFELD, Germany) and imaged with a spinning disk confocal microscope (ECLIPSE Ti-E, Nikon) with a Plan Apochromat 20× lens (N.A. = 0.75, WD = 1.0 mm) or an oil immersion objective lens Plan Apochromat 60× (N.A. = 1.40) and a Neo sCMOS camera (Andor Technology) using NIS-Elements software. For the measurement of neuronal ultrastructure integrity, Thy1-EYFP transgenic mouse brains were cut to a thickness of 50-μm and imaged at the same position before and after clearing using an oil immersion objective lens Plan Apochromat 60× (N.A. = 1.40). Then the acquired images were transferred to Fiji, and the same ROIs around cell bodies and dendrite segments before and after clearing were cropped, and aligned via rigid registration. For the measurement of the tortuosity of the dendrite segments, the straight length distance (D) between two ends of dendrite and actual length (d) were measured and analyzed as the ratio between them (d/D). For SSIM analysis of cell bodies, the registered images of

before and after clearing were binarized and analyzed for SSIM index using the ICY program (Institute Pasteur, France) with default parameters.

3) Light-sheet fluorescence microscope: Fluorescence images of cleared tissue including the whole brain and other whole organs were visualized with a light-sheet microscope (Ultra microscope, LaVision BioTec, Germany) equipped with a 2× objective lens (MVPLAPO 2XC, NA = 0.5, WD = 10 mm) and a dipping cap. Light in the thin plane was illuminated from both sides of the sample. Light-sheet images were collected with a Neo sCMOS camera (Andor Technology) using the ImSpector program (LaVision BioTec), stored as TIFF image stacks. For the entire imaging of whole organs, the z-stacks of 4 or 5 μm intervals were obtained with LSFM, and the IMARIS software program was used for 3D reconstruction and rendering. The voxel size of each 3D image was determined by calculating the image size of the raw data.

3D LSFM images of DiI-labeled organs (intestine, and spleen) were imported into IMARIS for 3D surface rendering. The background was subtracted by setting the value of the diameter of the largest sphere which fits into the object to 10 μm. The threshold was adjusted to achieve the best fit of the surface of the vasculature of interest and a volume filter was applied to exclude non-specific signals. Then, 3D rendered images were converted into TIFF files, transferred to NIS-element software, and color-coded according to the depth.

**Statistics and reproducibility.** Statistical comparisons and graph construction were performed using Prism 8.0 (GraphPad Software, Inc.). Relevant p-values were shown in figure legends, and data were presented as mean ± SD unless otherwise indicated. All analyses using mice and rats were performed in at least three independent experiments. The number of independent experiments was; six for tissue size and transparency measurements (200-μm, 1-mm thick rat samples); three for whole mouse brain, preservation of endogenous fluorescence and tissue integrity measurement, SNR measurement, and DiI labeled with NTN kidney. The normality of data distribution was checked with the Shapiro-Wilk test. Student's two-sample t-test was performed for comparison of two groups. For multiple group comparisons, ordinary one-way ANOVA followed by Tukey's multiple comparison test was performed. Statistical significance was shown as followed; ***$p < 0.001$, **$p < 0.01$, *$p < 0.05$.

**Reporting summary.** Further information on research design is available in the Nature Research Reporting Summary linked to this article.

## Data availability

The complete datasets including replicated measurement and analysis during the current experiment are available from the corresponding author on reasonable request. All source data for the graphs and charts are in Supplementary Data 1.

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

## Acknowledgements

We thank Dr. Chang Man Ha of Korea Brain Research Institute for providing transgenic mice samples. This study was funded by the Brain Research Program (NRF-2017M3C7A1044958) to S.C. through the National Research Foundation of Korea, Republic of Korea. The research was also supported by the Education and Research Encouragement Fund of SNUH (800-2020-0296).

## Author contributions

K.K., S.S.H., and S.C. designed the experiment. K.K., M.N., E.C., and K.O. performed the experiments. K.K., E.C. S.S.H., and S.C. analyzed the data. K.K. and S.C. prepared the manuscript.

## Competing interests

S.C. is the founder of and a shareholder in Crayon Technologies, Inc, South Korea. The remaining authors declare no competing interests.
