## [Peer Review File · Communications Biology]

Reviewers' comments:

Reviewer #1 (Remarks to the Author):

The manuscript titled "Optical clearing method to preserve maximum endogenous fluorescence and sample size for three-dimensional volume imaging" introduces a new clearing method named OptiMuS, a combination of the reagents: urea, iohexol, and sorbitol, to perform fast organ clearing. The authors applied the clearing to different mouse organs: brain, kidney, spleen, and intestine in combination with various labeling, both endogenous (YFP) and exogenous (immunostaining and lipophilic tracer), followed by confocal and light sheet fluorescence volumetric imaging. Finally, as an application, a comparison between the glomeruli's size in normal and murine nephrotoxic nephritis affected mice was performed using the Dexplorer tool, a machine-learning-based 3D morphology analysis program developed by the authors.

This is a comprehensive work, nevertheless, there are some major aspects that I would like to be analyzed:

1. The method section lacks important and essential information:

- Sample preparation and OptiMus clearing should be described in detail to permit other laboratories to replicate the protocol. All the steps should be indicated not only the solutions used. For example, you can not say "washed with PBS before clearing", you must indicate the time of washing, the quantity of the solution, the number of washing, and the temperature.
- Quantitative analysis should be described in detail as well. For example: in sentences like "The contour of the brain sample was extracted with ImageJ/Fiji program" and "The mean intensity of neuronal cell bodies ($n = 30 - 40$) and surrounding background intensity per image were measured using Fiji", there is missing information like how were they extracted and measured (e.g. Manually)? For how many samples (in the first case)?
- In the "Measurement of SNR." the acquisition of the images must be explicit: images were taken with what? Confocal? With which objective? Power of illumination? Did you change the pw in dependence of the depth? Which images were analyzed with Fiji? MIP? Single images at different depths? How was the dendrites' intensity selected? Manually? Segmenting the dendrites or some randomly px? How many?
- In the "NTN induction" and "Glomerulus classification and analysis": how many mice you analyzed? How many images for each sample? How many glomeruli?
- Even if the DXplorer was already published you must briefly explain how to use it. And if the DXplorer is available somewhere (i.e. GitHub) please provide the link.
- In "Statistics and reproducibility" are missing the number of the sample analyzed ("different samples" is not informative enough).

2. In the result section:

- The characterization of the OptiMus clearing is well performed, however, there is no mention on the Refractive Index (RI) of the final solution. Knowing the RI is important to be able to match it during the imaging in order to avoid aberration that lowers the quality of the image. Please measure and provide the RI of the final solution. Moreover, provide the information of the mounting you used for imaging, did you use glass or quartz coverslip/chambers?
- Authors claim that the OptiMus clearing does not cause tissue deformation, however, is not enough to provide the measurement of the mesoscopic changes (or preservation) of the tissue (as demonstrated in figure 1), high-resolution images should be provided. Indeed, as shown by Ke et al. 2013 Nat. Neurosc., Urea can introduce some deformation to the ultrastructure of dendrites and axons. Authors should provide high-resolution images that show the integrity of the fibers and not only of the cell body as shown in figure 2, maybe performing the confocal microscope imaging using a 60x or a 100x immersion objective.
- Concerning the mouse organ immunostaining in the result section there is a mention of a "SDS-based clearing method" that is not described in the method section, please include it. Moreover, it is not clear if the immunostaining is performed on the whole organ or if it is performed on some smaller block, please specify it.
- In the DiI staining chapter, there is a mention of a three-dimensional tracing of the vessels applicable to the images as shown in figure 4c-f. However, there is no explanation on how it was performed in the method section, please include it.

- Better quantitative description of the accuracy of the DXplorer for glomeruli segmentation should be provided. The main issue I have with it is that the software was developed for dendritic spines classification and accuracy could be quite variable with another structure. Looking at figure 5 it should be possible to quantify the accuracy between the automatic segmentation performed by the software and manual segmentation of the glomeruli and give the precision and recall rate of the tool.
- Finally, it would be good to discuss the impact of the software accuracy on the results presented in Figure 5f to verify if the accuracy of the software is sufficient to verify the differences observed between the control and the diseased kidney.

Minor:

- 2,2'-thiodiethanol (TDE) is miscible with water and does not produce endogenous fluorescence proteins quenching (as demonstrate by Aoyagi, Y. et al 2015 and Costantini et al 2015). To avoid confusion in the introduction please remove it from the organic section and put it in the aqueous section.
- The heart is another important organ that is really difficult to clear, have you tried OptiMus clearing on it?
- The 3D volume renderings of LSFM imaging are interesting, however, horizontal plane images are more informative to appreciate the contrast reached with the clearing, could you include it in figure 3?
- In the transcordial perfusion method section I think you meant snipping the atrium, not the aorta. Please check it.
- In the Statistics and reproducibility method section, you wrote rats, but there aren't other mentions of experiments on rats in the text, please check it.

Reviewer #2 (Remarks to the Author):

Chang et al. reported an optical clearing method, OptiMuS, which preserves endogenous fluorescence and sample sizes. The reported method was compared to six previously published clearing methods in terms of processing time, size changes of tissues, transmittance in the range of 400 to 800 nm and fluorescence retention of EYFP. Chang et al. claimed that OptiMuS had the best overall performance. They also demonstrated using OptiMuS to clear thick tissues, visualize neural networks and vascular structures and for a potential application to diagnose pathological status of kidneys.

Major issues:

1. Different clearing methods were tested on 1-mm thick rat brain tissues and the results were compared. In the Method section, authors claimed using ScaleS0 – S4 to process the brain slices. However, in the original publication (reference 14), ScaleS0 – S4 was designed for clearing cerebral hemisphere and ScaleSQ(0) was used for clearing brain slices. Similarly, in the Method section of FOCM (reference 25), it was suggested clearing 20- to 300-um slices for 1-5 min. Whereas, the authors claimed immersion of tissues in FOCM solution for 40 min (which could lead to the significant discrepancies between the manuscript and the original publication). Alterations of the published protocols need to be justified in the manuscript. Otherwise, the comparison among different clearing techniques will be significantly compromised and remain questionable.
2. In addition to point 1, the ingredients of ScaleSQ(0) are 22.5 (w/v)% D-sorbitol and 9.1 M Urea. The recipe used in the manuscript only changed the concentration of the ingredients (10 % (w/v) D-sorbitol and 4 M urea). It was reported that clearing a 1-mm-thick brain slice took 1-2 h with ScaleSQ(0) and 1.5 h with OptiMuS solution. The significance of the study is compromised although the authors demonstrated an potential application of OptiMuS to predict the pathological status of kidneys.
3. In figure 1, seven clearing methods were compared. In figure 2, MACS is missing for analysis of fluorescence retention. In Supp. Table 1, CLARITY is missing for overall comparison.

Reviewer #3 (Remarks to the Author):

In the manuscript entitled "Optical clearing method to preserve maximum endogenous fluorescence and sample size for three-dimensional volume imaging", Kim et al. proposed a simple single-step solution that tries to meet all needs. However, the results showed in the manuscript are not enough to illustrate the advantages of this method. I have several concerns regarding the validity and significance of this 'perfect' approach as below:

Major concerns:

1. Validity

1) The authors named their method OptiMus (Optical clearing method to preserve Maximum endogenous fluorescence and sample Size for three-dimensional volume imaging), which emphasized its ability to preserve endogenous fluorescence and sample size. However, these two factors are susceptible to the specific conditions, such as the type of tissue or even the age of the animals.

2) Besides, the MACS and FOCM show the good ability of tissue maintenance in their papers. So there are some reasons why it fails to perform well in this manuscript. And it turns out that this manuscript simplified the MACS protocol to one step and got the FOCM formula wrong which missing the glycerol. Meanwhile, the results of the endogenous fluorescence preservation experiment are not reliable because they did not record the fluorescence signals at the same position.

2. Significance

The significance of this method or the innovation is not fully demonstrated. It seems it has fewer improvements compared to MACS and its applications are similar to MACS.

3. Applications

The applications showed in this manuscript cannot demonstrate the advantages of OptiMus. Are there any applications that are more suitable for this new clearing method than other clearing methods?

4. Others

1) In the abstract, the authors claim that OptiMuS could provide the best performance in all aspects. But in the manuscript, OptiMuS has only shown advantages in a few aspects. For example, only the 1-mm thick tissue transparency and clearing time have been quantitatively analyzed. However, the results of thinner tissue or whole organs such as whole brain have not been quantitatively discussed.

2) The authors declare that the application of Iohexol was limited in thick tissues. However, Iohexol was used in SeeDB2, a simple clearing method with an excellent performance in mouse embryo and whole brain.

Minor concerns:

1. RIMs should be RIMS.

2. The manuscript needs to be carefully checked for grammatical errors and inconsistencies.

Reviewer #1 (Remarks to the Author):

The manuscript titled “Optical clearing method to preserve maximum endogenous fluorescence and sample size for three-dimensional volume imaging” introduces a new clearing method named OptiMuS, a combination of the reagents: urea, iohexol, and sorbitol, to perform fast organ clearing. The authors applied the clearing to different mouse organs: brain, kidney, spleen, and intestine in combination with various labeling, both endogenous (YFP) and exogenous (immunostaining and lipophilic tracer), followed by confocal and light sheet fluorescence volumetric imaging. Finally, as an application, a comparison between the glomeruli’s size in normal and murine nephrotoxic nephritis affected mice was performed using the Dexplorer tool, a machine-learning-based 3D morphology analysis program developed by the authors.

This is a comprehensive work, nevertheless, there are some major aspects that I would like to be analyzed:

1. The method section lacks important and essential information:

- Sample preparation and OptiMus clearing should be described in detail to permit other laboratories to replicate the protocol. All the steps should be indicated not only the solutions used. For example, you can not say “washed with PBS before clearing”, you must indicate the time of washing, the quantity of the solution, the number of washing, and the temperature.

>> We thank the reviewer for pointing this out. In accordance with the comment, we described details regarding experimental procedures such as time, the quantity of solution, etc. in the revised manuscript. For example, ‘100 ml of PBS’, ‘50 ml of 4 % PFA’ and ‘the speed of the peristaltic pump (5 ml/min)’ were now described in the sample preparation section of the revised manuscript. Additionally, the sentence pointed out was modified as follows. “All harvested samples were immersed overnight in 4 % PFA at 4 °C on a rotator for post-fixation and then washed with 50 ml PBS in a 4 °C rotator for 1 hr to remove residual PFA before clearing.” (In page 20, Line 361-364)

2. Quantitative analysis should be described in detail as well. For example: in sentences like “The contour of the brain sample was extracted with ImageJ/Fiji program” and “The mean intensity of neuronal cell bodies (n = 30 - 40) and surrounding background intensity per image were measured using Fiji”, there is missing information like how were they extracted and measured (e.g Manually)? For how many samples (in the first case)?

>> According to the reviewer’s suggestion, a detailed explanation was provided in the ‘quantitative analysis’ section of the revised manuscript. We have modified the sentence as follows: “The contours of three brain samples were extracted using ‘outlines’ tool of the ImageJ/Fiji program after thresholding images” and “The mean intensity of neuronal cell bodies (n = 8-10) and the surrounding background areas were measured manually using the freehand selection tool in Fiji”. And we now described “we used 3 mouse or rat brain samples”

(In page 21, Line 377-378 and page 22, Line 399-401)

3. In the “Measurement of SNR.” the acquisition of the images must be explicit: images were taken with what? Confocal? With which objective? Power of illumination? Did you change the pw in dependence of the depth? Which images were analyzed with Fiji? MIP? Single images at different depths? How was the dendrites' intensity selected? Manually? Segmenting the dendrites or some randomly px? How many?

>> The images were taken with the equal laser intensity of 150 μ W regardless of depth with a confocal microscope with a Plan Apochromat 10 x objective lens (NA 0.5, WD 5.5 mm). Then, the acquired images were transferred to Fiji for further analysis. From the optical sectioned single image at each depth, we selected 5 dendrites randomly, drew a line perpendicular to the dendrite axis, and measured the fluorescence intensities of the dendrites and adjacent surrounding areas. From the surface to a depth of 280 μ m, we measured the SNR value at 20 μ m intervals, and then the SNR values at depths of 600, 800, and 1,000 μ m were measured and plotted. (In page 22, Line 404-412)

4. In the “NTN induction” and “Glomerulus classification and analysis”: how many mice you analyzed? How many images for each sample? How many glomeruli?

We thank the reviewer to point this out. we have discussed in the answer to #5 below.

5. Even if the DXplorer was already published you must briefly explain how to use it. And if the DXplorer is available somewhere (i.e GitHub) please provide the link.

>> The DiI-labeled whole mouse kidneys were imaged as z-stacks at 4 μ m intervals using a light-sheet fluorescence microscope and the obtained images were transferred to the DXplorer software. Then, 3D images were interpolated along the z-direction to reduce the anisotropic resolution of the data. Then, the glomerulus was extracted using the geodesic active contour method applied to the binarized image using Otsu's thresholding method. Using an ellipse-fitting scheme, each glomerulus was separated and volume-rendered. The final triangular meshes were generated by the marching cube method for each detected glomerulus segment, and the surface was smoothed using a curvature flow smoothing algorithm. Because of the acquisition artifact and noise of the LSFM image data, however, incorrect or blurred glomeruli can often be detected especially around the edges of imaging planes. Therefore, using the intuitive and interactive proofreading feature of DXplorer, we confined the analysis to the center of the kidney structure and also verified how well the glomeruli were detected by interacting with the 3D visualization, which finds a specific glomerulus through 3D interaction such as rotation, zooming, or panning. After verification, 10 glomeruli per kidney were randomly selected and analyzed.

To measure 3D features, three anchor points, i.e., Central base point, Centroid, and Tip, were automatically extracted from the triangular mesh representing the glomeruli. The orientation of the glomerulus was then set as the glomerulus direction from the Central base point to

Centroid. The vertices were divided into steps of a certain height along with the orientation of the glomerulus from the Central base point to Tip, and L was measured as the length of the curve connecting the center of the local vertices in each step. hMin and hMax were measured as the minimum and maximum diameters of the thickest part between Centroid and Tip. See ref (PMID: 34591770) for more details regarding DXplorer analysis. As requested by the reviewer, we provided GitHub links of DXplorer (https://github.com/hvcl/SpineAnalysis_public) in the 'Glomerulus classification and analysis' section of the Method. (In page 24, Line 452-474)

6. In “Statistics and reproducibility” are missing the number of the sample analyzed (“different samples” is not informative enough).

>> As the reviewer pointed out, we modified the sentence to include the details of the analysis. For example, we modified as follows: Relevant p-values were shown in figure legends, data were presented as mean \pm SD unless otherwise indicated. All analyses using mice and rats were performed in at least three independent experiments. The number of independent experiments was; 6 for tissue size and transparency measurements (1 mm thick rat samples); 3 for whole mouse brain, 200 μ m thick rat brain sample, preservation of endogenous fluorescence and tissue integrity measurement, SNR measurement, and DiI labeled with NTN kidney. (In page 28, Line 558-563)

In the result section:

7. The characterization of the OptiMus clearing is well performed, however, there is no mention on the Refractive Index (RI) of the final solution. Knowing the RI is important to be able to match it during the imaging in order to avoid aberration that lowers the quality of the image. Please measure and provide the RI of the final solution. Moreover, provide the information of the mounting you used for imaging, did you use glass or quartz coverslip/chambers?

>> As the reviewer pointed out, we now measured the refractive index (RI) of the final solution and included the information (Supplementary Table 1). (In page 7, Line 125-126)

And in the imaging part of the method section, how the sample was set and imaged according to the type of experiment was classified and described.

1) Upright confocal microscope: The sample was placed on a 35 mm Petri dish, immobilized on a 22 mm glass coverslip, immersed in OptiMuS, and imaged with an upright confocal microscope (Nikon C2Si, Japan) with a Plan-Apochromat 10 \times lens (NA = 0.5, WD = 5.5 mm) using NIS-Elements software (Nikon).’

2) Inverted confocal microscope: The sample was set on a coverglass (HSU-0101152, MARIENFELD, Germany) and imaged with a spinning disk confocal microscope (ECLIPSE Ti-E, Nikon) with a Plan Apochromat 20 \times lens (N.A.=0.75, WD= 1.0 mm) or an oil immersion objective lens Plan Apochromat 60 \times (N.A. =1.40) and a Neo sCMOS camera

(Andor Technology) using NIS-Elements software.

3) Light-sheet fluorescence microscope: The whole brain and other whole organs were visualized with a light-sheet microscope (Ultra microscope, LaVision BioTec, Germany) equipped with a 2x objective lens (MVPLAPO 2XC, NA= 0.5, WD=10 mm) and a dipping cap. The samples were immobilized in the sample chamber and then immersed in OptiMuS solution for imaging. The light in the thin plane was illuminated from both sides through the sample. Light-sheet images were collected through the ImSpector program (LaVision BioTec). (In page 26-28, Line 522-555)

8. Authors claim that the OptiMus clearing does not cause tissue deformation, however, is not enough to provide the measurement of the mesoscopic changes (or preservation) of the tissue (as demonstrated in figure 1), high-resolution images should be provided. Indeed, as shown by Ke et al .2013 Nat. Neurosc., Urea can introduce some deformation to the ultrastructure of dendrites and axons. Authors should provide high-resolution images that show the integrity of the fibers and not only of the cell body as shown in figure 2, maybe performing the confocal microscope imaging using a 60x or a 100x immersion objective.

>> As the reviewer suggested, we performed a 60X high-resolution confocal imaging before and after clearing to show that OptiMuS did not cause the deformation of the ultrastructure (Supplementary Fig 2). (In page 8, Line 136-139)

We measured the tortuosity of the same dendrite segments before and after clearing and found no difference between them (Supplementary Fig 2 c,e). We also measured SSIM (structure similarity index measure) of the soma between before and after clearing and confirmed that there was no significant structural change (Supplementary Fig 2 b,d). (In page8, Line 136-139; in supplementary information page 3, Line 24-28)

9. Concerning the mouse organ immunostaining in the result section there is a mention of a “SDS-based clearing method” that is not described in the method section, please include it. Moreover, it is not clear if the immunostaining is performed on the whole organ or if it is performed on some smaller block, please specify it.

>> Thank you for pointing this out. To avoid confusion, we deleted ‘SDS-based clearing’ and replaced it with CLARITY. In this regard, the method section was modified accordingly. “...brains (700- μ m thick block) and intestines (whole organ) were cleared with CLARITY protocol and then immunostained”. Information on the sample size was now described in the ‘immunostaining’ part of the method. (In page 22, line 414- 418)

10. In the DiI staining chapter, there is a mention of a three-dimensional tracing of the vessels applicable to the images as shown in figure 4c-f. However, there is no explanation on how it was performed in the method section, please include it.

>> We now described the details of the three-dimensional tracing of the vessels in the ‘imaging’ section of the method.

3D LSFM images of DiI-labeled organs (brain, intestine, and spleen) were imported into IMARIS for 3D surface rendering. The background was subtracted by setting the value of the diameter of the largest sphere which fits into the object to 10 μm . Threshold was adjusted to achieve the best fit of the surface of the vasculature of interest and a volume filter was applied to exclude non-specific signals. Then, 3D rendered images were converted into TIFF files, transferred to NIS-element software, and color-coded according to the depth. (In page 27-28, Line 550-555)

11. Better quantitative description of the accuracy of the DXplorer for glomeruli segmentation should be provided. The main issue I have with it is that the software was developed for dendritic spines classification and accuracy could be quite variable with another structure. Looking at figure 5 it should be possible to quantify the accuracy between the automatic segmentation performed by the software and manual segmentation of the glomeruli and give the precision and recall rate of the tool.

>> Although DXplorer was originally developed to analyze dendritic spine morphology, due to morphological similarity between dendritic spines and glomeruli, we expected DXplorer could be applied to analyze glomeruli structure since algorithms for feature rendering and detection are not limited to a specific structure but can be applied to any structure with a round-shape head and a connecting neck. Since there is about a 20-fold difference in size between the glomerulus and the spine, we reduced the pixel dimension of the glomerulus to the size of the spine before loading the image into DXplorer, which was later reflected in the analysis.

As the reviewer indicated, because of the acquisition artifact and noise of the LSFM image data, however, incorrect or blurred glomeruli can often be detected especially around the edges of imaging planes. Therefore, it is essential for users to proofread the detection. For the intuitive and interactive proofreading feature of DXplorer, we verified how well the glomeruli were detected by interacting with the 3D visualization, which finds or focuses on a specific glomerulus through 3D interaction such as rotation, zooming, or panning.

In addition, to check the accuracy of DXplorer's automatic detection and feature extraction, we manually isolated glomeruli using the ‘3D crop’ function of Imaris and imported them to Meshlab program to construct 3D meshed images, and then measured the volume and maximum head/neck diameter.

Then, we compared these values with DXplorer’s and found that there is no significant difference between them (Supplementary Fig 8). (In page 11 Line 222-226)

12. Finally, it would be good to discuss the impact of the software accuracy on the results presented in Figure 5f to verify if the accuracy of the software is sufficient to verify the

differences observed between the control and the diseased kidney.

>> We are confident that the performance of the software is accurate if the acquired images are of good quality. Indeed, the bottleneck of accurate detection and analysis by DXplorer is not the performance of the software but 3D image acquisition. Again, algorithms for feature rendering and detection of DXplorer are not limited to a specific structure but can be applied to any structure that protrudes from the base structure such as a head and connecting neck. The glomerulus of the kidney was one of them.

Although we didn't perform further analysis in this study, since the updated version of DXplorer (PMID: 34591770) can provide 10 different features including volume, hMax, and hMin, one can obtain more insightful data regarding the 3D structures of interest. More importantly, DXplorer can be trained to perform automatic machine-learning-based feature classification. Thus, using this feature, different shapes of diseased kidneys can be objectively classified, and these data can be used to monitor the progress of the diseases. For example, the kidney of the NTN disease model undergoes a series of progressions over time, such as mesangial expansion and fibrosis, which can be classified by DXplorer to objectively diagnose the disease state. It is of future interest although certainly requires extensive further studies. (In page 13, Line 264-275)

Minor:

13. -2,2'-thiodiethanol (TDE) is miscible with water and does not produce endogenous fluorescence proteins quenching (as demonstrate by Aoyagi, Y. et al 2015 and Costantini et al 2015). To avoid confusion in the introduction please remove it from the organic section and put it in the aqueous section.

>> We apologize for the confusion. -2,2'-thiodiethano (TDE) was transferred to the aqueous section. (In page 4, Line 66-67)

14. The heart is another important organ that is really difficult to clear, have you tried OptiMus clearing on it?

>> As requested by the reviewer, we added mouse and rat heart clearing data in Supplementary figure 5. (In supplementary information page 6, Line 39-45)

15. The 3D volume renderings of LSFM imaging are interesting, however, horizontal plane images are more informative to appreciate the contrast reached with the clearing, could you include it in figure 3?

>> As the reviewer mentioned, we added a horizontal plane image of the 3D Thy1-EYFP brain in Figure 3. (In page 17, Line 319-320)

16. In the transcordial perfusion method section I think you meant snipping the atrium, not

the aorta. Please check it.

>> We apologize for a careless mistake. As the reviewer pointed out, we corrected the word aorta to the atrium. (In page 23, Line 440-441)

17. In the Statistics and reproducibility method section, you wrote rats, but there aren't other mentions of experiments on rats in the text, please check it.

>> We modified all figure legends separately for rat and mouse. (In page 28, Line 557-566)

Reviewer #2 (Remarks to the Author):

Chang et al. reported an optical clearing method, OptiMuS, which preserves endogenous fluorescence and sample sizes. The reported method was compared to six previously published clearing methods in terms of processing time, size changes of tissues, transmittance in the range of 400 to 800 nm and fluorescence retention of EYFP. Chang et al. claimed that OptiMuS had the best overall performance. They also demonstrated using OptiMuS to clear thick tissues, visualize neural networks and vascular structures and for a potential application to diagnose pathological status of kidneys.

Major issues:

1. Different clearing methods were tested on 1-mm thick rat brain tissues and the results were compared. In the Method section, authors claimed using ScaleS0 – S4 to process the brain slices. However, in the original publication (reference 14), ScaleS0 – S4 was designed for clearing cerebral hemisphere and ScaleSQ(0) was used for clearing brain slices.

Similarly, in the Method section of FOCM (reference 25), it was suggested clearing 20- to 300-um slices for 1-5 min. Whereas, the authors claimed immersion of tissues in FOCM solution for 40 min (which could lead to the significant discrepancies between the manuscript and the original publication).

Alterations of the published protocols need to be justified in the manuscript. Otherwise, the comparison among different clearing techniques will be significantly compromised and remain questionable.

>> We thank the reviewer to point this out. Regarding the issue related to ScaleSQ(0), we have discussed in the answer to #2 below.

FOCM was designed to clear relatively thin samples (20-300 μm) within a short period of time. In Figure 1 of our experiment, however, the thickness of the sample was 1-mm (Fig. 1c). Since there was no information on the time to clear thicker samples in the original paper of FOCM, we arbitrarily set the time by estimating the time according to the sample thickness. Indeed, when we cleared the 1-mm thick sample with FOCM for just 5 min, the transparency obtained was much lower compared to other methods (Fig.1) (In page 12 Line 246-247)

2. In addition to point 1, the ingredients of ScaleSQ(0) are 22.5 (w/v)% D-sorbitol and 9.1 M Urea. The recipe used in the manuscript only changed the concentration of the ingredients (10 % (w/v) D-sorbitol and 4 M urea). It was reported that clearing a 1-mm-thick brain slice took 1-2 h with ScaleSQ(0) and 1.5 h with OptiMuS solution. The significance of the study is compromised although the authors demonstrated an potential application of OptiMuS to predict the pathological status of kidneys.

>> We thank the reviewer to point this out. Indeed, the difference between OptiMUS and ScaleSQ is not only using different sorbitol and urea concentrations but also combining a high concentration of iohexol. (In page 7, Line 119-126)

As the reviewer indicated, compared to Scale SQ, OptiMuS resulted in no significant improvement in clearing time, but it is noteworthy that, according to the original Scale paper, tissue size increased by ~ 20% after Scale SQ clearing, which we also observed in Fig. 1c. Thus, a high concentration of urea (9.1 M) can make the sample transparent but significant size change is inevitable. ScaleSQ data were newly added to Figure 1 for comparative analysis. (In page 7, Line 131)

We proved the importance of using three compounds together to ensure maximum transparency and size maintenance (Supplementary Fig 1). Through vigorous screening, we found that using a moderate concentration of Urea (4 M) with 10% D-sorbitol was the optimal concentration for size maintenance along with considerable sample transparency (58 % transparency, Supplementary Fig.1 a-c). When we combined iohexol (75%), we were able to achieve higher transparency (> 79 %) without sample size change (Supplementary Fig.1c), thus positioning OptiMuS as the newly formulated and optimized method that ensures the maximum transparency along with size maintenance (Fig.1e). (In page 7, Line 119-126)

3. In figure 1, seven clearing methods were compared. In figure 2, MACS is missing for analysis of fluorescence retention. In Supp. Table 1, CLARITY is missing for overall comparison.

>> MACS analysis was added to the fluorescence retention data in Figure 2, and CLARITY was added in Table 2. (In page 8, Line 149-151; In supplementary information page 11)

Reviewer #3 (Remarks to the Author):

In the manuscript entitled “Optical clearing method to preserve maximum endogenous fluorescence and sample size for three-dimensional volume imaging”, Kim et al. proposed a simple single-step solution that tries to meet all needs. However, the results showed in the manuscript are not enough to illustrate the advantages of this method. I have several concerns regarding the validity and significance of this ‘perfect’ approach as below:

Major concerns:

1. Validity

1) The authors named their method OptiMus (Optical clearing method to preserve Maximum endogenous fluorescence and sample Size for three-dimensional volume imaging), which emphasized its ability to preserve endogenous fluorescence and sample size. However, these two factors are susceptible to the specific conditions, such as the type of tissue or even the age of the animals.

>> We fully agree with the reviewers that size and endogenous fluorescence retention are susceptible to a variety of situations, however, it is not practically possible to test the validity of the method in all possible situations. Indeed, this applies not only to OptiMus but also to other clearing methods available.

Nevertheless, in this study, we clearly showed that OptiMus outperformed other methods for clearing the rat/mouse brain tissues. We also showed that OptiMus can be applied to clear other organs such as the liver, kidney, spleen, heart, intestine, lung, and spinal cord (Supplementary Fig 5). (In page 8, Line 143-146)

We, however, in response to the concerns of the reviewer, omitted Maximum, thus OptiMus is now “*Optimized single-step optical clearing Method that preserves fluorescence and Size of the sample for 3D volume imaging*”. (In page 5, Line 93-94)

2) Besides, the MACS and FOCM show the good ability of tissue maintenance in their papers. So there are some reasons why it fails to perform well in this manuscript. And it turns out that this manuscript simplified the MACS protocol to one step and got the FOCM formula wrong which missing the glycerol. Meanwhile, the results of the endogenous fluorescence preservation experiment are not reliable because they did not record the fluorescence signals at the same position.

>> Currently, we do not know why MACS and FOCM did not perform as published. We, however, repeated the experiments several times and are confident of the results we obtained.

The reason we used one solution (R1) for MACS clearing is that in the original MACS paper (PMID: 32328422) the author used only one solution to clear 1-mm thick brain tissue (Figure S1 in MACS paper), and we just followed their protocol. Anyhow, in this revision, we performed MACS clearing using 3 solutions protocol (R0, R1, R2) and found that the result was similar to using one solution (Fig. 1).

In the case of FOCM, we indeed used glycerol but mistakenly failed to mention that in the Method. We apologized for the careless mistake. To clarify this issue, we have repeated FOCM clearing experiment (with glycerol) in this revision and again found that there was a decrease in size as well as the disappearance of endogenous fluorescence (Fig 1,2a). (In page 12, Line 243-250)

Following the reviewer's suggestion, in this revision, endogenous fluorescence preservation experiments over time were performed with each method at the same location (Fig 2a). (In page 8, Line 149-151)

2. Significance

The significance of this method or the innovation is not fully demonstrated. It seems it has fewer improvements compared to MACS and its applications are similar to MACS.

>> Although we agree with the reviewer that the application of OptiMuS seems similar to MACS, since the purpose and applications of most clearing methods are more or less identical, we believe that this applies not only to OptiMuS but also to other clearing methods.

OptiMuS provides a single-step solution for simple, fast, and versatile optical clearing method to obtain high tissue transparency with minimum structural changes.

Compared to MACS, 1) OptiMuS is the simple protocol that uses only 1 clearing solution while MACS requires sequential incubation in three different compositions of MXDA and sorbitol. 2) OptiMuS shows higher transparency than MACS while MACS shows only moderate levels of transparency in the visible wavelength range, 3) OptiMuS preserves the sample size while MACS results in significant shrinkage of the sample. (In page 12, Line 243-246)

Again, the purpose of tissue clearing is straightforward; to clear tissues to see deep inside. Therefore, we believe that higher transparency, better sample size preservation, and improvements in simple procedures in tissue removal methodologies are significant improvements.

3. Applications

The applications showed in this manuscript cannot demonstrate the advantages of OptiMus. Are there any applications that are more suitable for this new clearing method than other clearing methods?

>> As mentioned above, most of the clearing methods have almost the same purpose and applications. The advantage of OptiMuS is that it achieves higher tissue transparency with minimum size change in a single-step simple procedure thus overcoming the shortcomings of other clearing techniques, making it a better alternative for a variety of applications requiring tissue clearing.

Due to its simple procedure and excellent size preservation, OptiMuS can be combined with

other biomedical imaging disciplines. For example, optical coherence tomography (OCT) which is used for non-invasive label-free volumetric histopathology analysis can be used with OptiMuS. OCT is based on low-coherence interferometry and is employed in diverse applications, including diagnostic medicine such as in ophthalmology, optometry, and dermatology to improve diagnosis. The application of OCT is, however, limited to depths of a few micro-meter from the surface thus it requires thousands of slices to construct an entire organ image. Although yet preliminary, we have started to collaborate with the OCT research group by taking advantage of the fact that OptiMuS preserves the size of structures while achieving tissue transparency. With OptiMuS if the whole organ is cleared to some extent, cut into several slices, and imaged with OCT, we could reliably register several OCT images together to construct whole 3D images due to OptiMuS's excellent size preservation. Thus, we found that combining OptiMuS with OCT markedly increases the penetration depth and speed of OCT imaging and thus could get label-free images of various entire organs using OptiMuS-based OCT within a few minutes.

We now explicitly discussed this point in the revision. (In page 13-14, Line 277-290)

4. Others

1) In the abstract, the authors claim that OptiMuS could provide the best performance in all aspects. But in the manuscript, OptiMuS has only shown advantages in a few aspects. For example, only the 1-mm thick tissue transparency and clearing time have been quantitatively analyzed. However, the results of thinner tissue or whole organs such as the whole brain have not been quantitatively discussed.

>> As suggested by the reviewer, in addition to 1-mm thick sample, we now included the quantitative analysis data of 200 μm thick rat brain and whole mouse brain. All data were presented in the revised manuscript (Supplementary figure 3,4). (In page 8, Line 141-143)

2) The authors declare that the application of Iohexol was limited in thick tissues. However, Iohexol was used in SeeDB2, a simple clearing method with an excellent performance in mouse embryo and whole brain.

>> Iohexol solution has previously been reported to improve the transparency of thin cornea samples (PMID: 3304411). However, iohexol alone could not efficiently clear thick brain samples due to inefficient penetration of iohexol into thick tissues. Thus, in the SeeDB2 paper (PMID: 26972009), the author used 2% saponin as a detergent to facilitate iohexol clearing.

As the reviewer indicated, SeeDB2 was shown to successfully clear mouse embryo or newborn mouse brain. According to data from a recent paper (PMID: 32328422), however, it resulted in poor clearing performance in adult whole mouse brain. We also showed that OptiMuS significantly outperformed SeeDB2 in tissue transparency (1 mm thick rat sample SeeDB2 25 %, OptiMuS 70.3 % at 600 nm in Figure 1d; 200 μm thick rat sample SeeDB2

77 %, OptiMuS 98 % in Supplementary Figure 3c). (In page 8, Line 141-143; supplementary information page 4-5)

Anyhow, to avoid unnecessary confusion, we rephrased the sentence appositely.

Minor concerns:

1. RIMs should be RIMS.

>> Following the reviewer's comment, we have changed RIMs to RIMS. (In page 4, Line 71)

2. The manuscript needs to be carefully checked for grammatical errors and inconsistencies.

>> As pointed out by the reviewer, grammatical errors and inconsistencies of the manuscript were corrected.

Reviewers' comments:

Reviewer #1 (Remarks to the Author):

The manuscript has been greatly improved in this submission thanks to the modification and the additional analysis made. I, therefore, recommend publishing the manuscript.

Minor:

I suggest adding also ref 21 when you are referring to TDE (line 67) since the article presents a clearing approach that used TDE in combination with Two-photon Fluorescence Microscopy in addition to the CLARITY/TDE method for Light Sheet Microscopy.

Reviewer #2 (Remarks to the Author):

The authors successfully addressed issue 2 and 3. However, there are still two concerns about the revised manuscript.

Page and line numbers below are based on the word version of the revised and mark-up manuscript.

1. Page 13 line 282, OCT can image tissues with image depth of several hundred micrometers [Kut, 2015, Huang, 2017]. Combining clearing and OCT is an interesting topic. Please rephrase the whole paragraph.

Kut, Carmen, et al. "Detection of human brain cancer infiltration ex vivo and in vivo using quantitative optical coherence tomography." *Science translational medicine* 7.292 (2015): 292ra100-292ra100.

Huang, Yongyang, et al. "Optical coherence tomography detects necrotic regions and volumetrically quantifies multicellular tumor spheroids." *Cancer research* 77.21 (2017): 6011-6020.

2. In the method section, authors clarified how the clearing time was determined for CLARITY (line 494, "...so that the sample becomes transparent.") and MACS (line 521, "...followed the original paper protocol."). Please clarify for all other clearing methods.

reviewers' comments:

Reviewer #1 (Remarks to the Author):

The manuscript has been greatly improved in this submission thanks to the modification and the additional analysis made. I, therefore, recommend publishing the manuscript.

Minor:

I suggest adding also ref 21 when you are referring to TDE (line 67) since the article presents a clearing approach that used TDE in combination with Two-photon Fluorescence Microscopy in addition to the CLARITY/TDE method for Light Sheet Microscopy.

>> We sincerely thank the reviewer for recommending the publication of the manuscript. As suggested by the reviewer we have added ref 21 when referring to TDE (In page 4, Line 66).

Reviewer #2 (Remarks to the Author):

The authors successfully addressed issue 2 and 3. However, there are still two concerns about the revised manuscript. Page and line numbers below are based on the word version of the revised and mark-up manuscript.

1. Page 13 line 282, OCT can image tissues with image depth of several hundred micrometers [Kut, 2015, Huang, 2017]. Combining clearing and OCT is an interesting topic. Please rephrase the whole paragraph.

Kut, Carmen, et al. "Detection of human brain cancer infiltration ex vivo and in vivo using quantitative optical coherence tomography." *Science translational medicine* 7.292 (2015): 292ra100-292ra100.

Huang, Yongyang, et al. "Optical coherence tomography detects necrotic regions and volumetrically quantifies multicellular tumor spheroids." *Cancer research* 77.21 (2017): 6011-6020.

>> We thank for the reviewer's kind suggestion. As suggested by the reviewer, in consultation with our OCT colleague, the paragraph was revised as follows with appropriate references.

"OptiMuS would be applicable to staining-free optical imaging modalities such as optical coherence tomography (OCT) rather than optical microscopes based on the fluorescent contrast. Since OCT uses near-infrared light and endogenous contrast, it offers deep tissue morphology over the range of several hundred micrometers [1-2]. However, the imaging depth of OCT is still limited for visualizing diagnostic features over large areas or volumetric contexts. Although yet preliminary, we found that OptiMuS could further enhance OCT imaging depth while showing a similar tendency to the previous work [3]. Moreover, the technical integration of OptiMuS and OCT would be well suited for the need for fast 3D histopathology, as it can exceptionally preserve the size of the tissue and can reduce scattering coefficient within a few minutes [4]. Taken together, a fully engineered imaging protocol would allow the investigation of 3D anatomical alterations associated with various diseases, and provide a paradigm shift towards digital histopathology." (In page 14, Line 293-303)

[1] Kut, Carmen, et al. "Detection of human brain cancer infiltration ex vivo and in vivo using quantitative optical coherence tomography." *Science translational medicine* 7.292 (2015)

[2] Huang, Yongyang, et al. "Optical coherence tomography detects necrotic regions and volumetrically quantifies multicellular tumor spheroids." *Cancer research* 77.21 (2017)

[3] Kwangyeol Baek, et al. "Quantitative assessment of regional variation in tissue clearing efficiency using optical coherence tomography (OCT) and magnetic resonance imaging (MRI): A feasibility study." *Scientific Reports*, vol. 9, pp. 2923, (2019)

[4] Eunjung Min, et al. "Serial optical coherence microscopy for label-free volumetric histopathology." *Scientific Reports*, vol. 10, pp. 6711, (2020)

2. In the method section, authors clarified how the clearing time was determined for CLARITY (line 494, "...so that the sample becomes transparent.") and MACS (line 521, "...followed the original paper protocol."). Please clarify for all other clearing methods.

>> As suggested by the reviewer, we clarified how the clearing time was determined for all other clearing methods. Indeed, the clearing time for each thickness of the tissue was determined according to the corresponding original report (In page 25-27, Line 506-563).

Reviewer #3

The manuscript by Kim et al. proposed an optimized single-step optical clearing solution for 3D volume imaging of biological structures named OptiMuS. However, I still have several concerns regarding the validity and significance of this optical clearing solution as below.

Major concerns:

1. Compared with other multi-step aqueous-based clearing methods, like CUBIC L/R or PACT, how to achieve better transparency without a specialized lipid-removal step or a higher RI? What is the most efficient clearing solution in OptiMuS? Does this method have its limitation when considering the simple combination of iohexol, urea, and sorbitol?

>> As the reviewer indicated, passive clearing by simple immersion in high refractive index solution renders tissues transparent although not as much clear as other delipidating-RI matching methods (**Review comment table 1**). But, OptiMuS overcame this limitation by combining iohexol (high RI solution) in conjunction with urea-mediated hydration. Urea is known to penetrate, denature and thus hydrate even the hydrophobic regions of high refractive index proteins (Hua et al., 2008). Urea, however, at its high concentration, causes sample deformation. We, thus formulated OptiMuS by combining D-sorbitol to control size expansion. Therefore, all three reagents are essential for optimal clearing performance of OptiMuS.

In this study, we proved that OptiMuS is optimized for clearing brain tissues. Since different organs have different compositions of lipids, proteins as well as the extracellular matrix, it should be optimized by carefully titrating the concentrations of each component for each organ for proper performance. Besides, since OptiMuS is a RI matching solution without a dilapidation process, it requires a separate delipidating process when performing post-immunostaining (In page 13, Line 268-274).

2. According to our experience, the transparency will be limited especially in the elder mouse brain. Meanwhile, the transmittance of the whole brain cannot reflect the real problems when imaging the whole brain. The signal quality deep inside the brain still has strong aberration which cannot meet the demands for analysis. For example, the results showed in Fig. 4a seem to lose the details of capillaries in the deep area.

>> As the reviewer mentioned, the clearing performance also highly depends on the sturdiness of tissues, and clearing of the aged brain is particularly challenging due to increased light scattering within the brain as well as lipid and protein accumulation in the sample.

We, however, would like to argue that **imaging is a completely separate issue from clearing**. We have used a commercial light-sheet microscope from LaVision BioTec. Although many improvements have been suggested in recent literature, current commercially available LSFM still suffers from the spreading of the light-sheet and its deterioration, thus severely limiting the capability of imaging large specimens. A conventional Gaussian illuminating beam should be focused on a micron-thick light sheet, it caused substantial diffraction and spreading of the light

sheet around the confocal region, leading to a blurry image around the focus. Besides, the light sheet is severally disturbed by the interference of various parts of the excitation beam after passing through the sample, leading to non-uniform illumination of the sample and resulting in artificial structures.

Surely, the aberration also can be caused by improper clearing, but we assure that this was not the case here. To prove that, we cleared the Dil-labeled whole mouse brain, obtained a 3D image with LSFM. Then, we halved the brain axially and imaged the inside of the brain (Figure 4a). Since the internal blood vessels in the transverse section are clearly visible, we confirmed that the blurring and aberration are not due to a clearing problem, but a limitation of LSFM imaging (In page 10, Line 191-194).

3. In the manuscript, there is a transparency contrast of 200 μ m, 1mm and the whole brain. However, it does not describe the process of tissue processing with different thicknesses. In Figure 1, supplementary Figure 3 and 4, the tissue deformation and transparency of comparison methods are far from the results reported. Therefore, I have doubts about the accuracy of the comparison of other methods in the manuscript.

>> We now have described the process of tissue clearing with different thicknesses in the method section in the revised manuscript (In page 25-27, Line 506-563).

The reviewer criticized that the degree of tissue deformation and transparency of comparison methods are far from the results reported. The main reason for the discrepancy in size deformation degree is due to the use of different measures (linear expansion vs. size change(area)). We used the size change measure which indicates the difference in the size in μm^2 ("area") while other methods used the "linear expansion" measure, which is the square root value of the area. Although we still believe that using size change in area is more intuitive and easier for the user to get a grip on the change of tissue size, to be consistent with the previous reports, we have replaced size change with linear expansion in this revision(In page 21, Line 395-398).

By replacing size change values with linear expansion values, we showed that the results were more or less comparable to previous reports except for FOCM and MACS. We still do not know why FOCM and MACS did not perform as reported though, we have repeated the experiments several times as indicated in the original papers but ended up with the same results (**Review comment table 2**).

Regarding transparency, most previous papers did not explicitly mention the transparency values, thus we could only roughly approximate the transparency from the figures presented in each paper. We found that our results are comparable to those reported (**Review comment table 3**).

4. In the method of FOCM (reference 27), zhu et al. has been claimed that the reagent ratio of whole brain or half brain is different from that of brain slices. However, the manuscript only used the reagent ratio for brain slices, but tried it on the whole brain.

>> We apologize for omitting the solution composition and clearing process depending on the thickness of the sample except for the 1 mm-thick sample. We indeed carefully followed the procedures described in the FOCM paper and changed the reagent composition accordingly. We now described in the Method section as follows. “For 200 μm and 1 mm-thick rat brain slices, FOCM solution consisting of 30 % (w/v) urea and 20 % (w/v) D-sorbitol in DMSO was used. For whole mouse brain clearing, FOCM solution consisting of 20 % (w/v) urea and 30 % (w/v) D-sorbitol in DMSO was used. After completely dissolution, 5% (w/v) glycerol was added and mixed thoroughly. Brain tissues were immersed in FOCM solution at RT for 2 min (200 μm), 5 min (1-mm), and 2 days (whole brain) to adjust RI, respectively.” (In page 27, Line 544-549)

5. Size preservation is very susceptible to specific conditions, and we believe every clearing method is optimized for its specific application. FOCM is designed for thin brain slices. But in the supplementary Fig. 3a, it shrinks very badly. There are some mistakes in reagents or operation. The details of this experiment are required. How long does it take? What is the refractive index of FOCM in this experiment?

>> We are aware that FOCM is designed for the rapid clearing of thin brain slices. Indeed, FOCM claimed to clear 300- μm -thick brain slices in less than 2 min.

In Supplementary Fig. 3, we compared the clearing performance of various clearing methods in 200 μm thick rat brain slices. Clearing time was set to 2 min in all methods. We have measured the RI value of the FOCM solution and found it to be 1.497 at 23.3 $^{\circ}\text{C}$, which is not different from the value (1.495) reported in the original paper.

We followed exactly the procedure reported in the FOCM paper and repeated the experiment several times but FOCM invariably caused shrinkage of 200 μm thick rat brain slices. The difference is that FOCM paper used the brain slices of the mouse (C57BL/6, 9 weeks old) while we used rat brain (Sprague-Dawley, 3 weeks old). In addition, the FOCM paper only reported the deformation value of the brain slice obtained from one hemisphere (Fig. 2 of FOCM paper, PMID : 31101714) and which depth of the coronal plane was used is not explicitly described. Since different anatomical brain regions are known to show somewhat differing clearing results, depending on the degree of myelination, we may suspect the reason for this discrepancy might be due to the different regions used or the existence of the corpus callosum, which is a large bundle of myriad myelinated fibers that connect the two brain hemispheres (In page 12-13, Line 251-260).

We, however, have no intention whatsoever to question the validity of FOCM clearing in this paper, and the exact cause of this discrepancy is currently unknown.

6. In the introduction, the author declares that OptiMuS could be applicable for clinically relevant samples. But there is nowhere in the manuscript showing the application of clinically relevant samples.

>> Although we haven't tested OptiMuS in clinically relevant samples in the current manuscript,

since we showed in Fig. 5 that OptiMuS applied to analyzed glomerulus structures from the kidney of the murine nephrotoxic nephritis (NTN) mouse disease model, an acute model of human glomerulonephritis and chronic kidney disease, we expect that it “could be” applicable for clinically relevant samples. But we agree with the reviewer on the point that it may be exaggerated expression without evidence, thus it was omitted in this revision (In page 6, Line 103).

7. In supplementary Figure 3, Coomassie Blue was used to help visualize. However, the compatibility of different optical clearing methods and Coomassie Blue is not consistent, which has a great impact on the organization. The transparency and organizational deformation calculated in this way are not credible.

>> Tissue transparency was measured **without Coomassie Blue staining**, and Coomassie staining was used only during taking pictures to measure the tissue size. Due to the high transparency of cleared tissues, we thought that Coomassie staining would help users to easily compare the size of tissues. To avoid unnecessary confusion, we have now replaced images that were not stained with Coomassie Blue (Supplementary Fig. 3).

8. For the two main points of this manuscript, deformation and fast (convenient), deformation can only be aimed at mouse brain, other organs are still difficult to maintain, and processing speed is not as fast as MACS.

>> We believe that the birth and recent surge in tissue clearing technology stems in large part from the need to visualize the 3D internal structures of the brain. Thus, most clearing techniques aim to clear brain tissues in the first place and the same is true for OptiMuS.

It is noteworthy, though, that different organs contain different compositions of lipids, proteins, and extracellular matrix, which mainly affect the performance of tissue clearing (PMID: 30143733), and this means that the composition of the clearing solution and the timeline should be optimized for the best performance in each organ.

The reviewer mentioned MACS’s performance in other organs, but a direct comparison between MACS and OptiMuS in other organs is not possible since quantitative information was not available in the MACS paper regarding the exact age of the animals from which each organ was isolated (only described as 8-12 weeks old), deformation degree after clearing, and the transparency obtained with MACS. We, thus, simply compared the clearing time of each organ with MACS and OptiMuS and summarized in a table (**Review comment table 4**).

Although, in principle, we agree with the reviewer’s point that the clearing performance in other organs seems not good enough compared to brain tissue, considering its excellent performance in brain tissue, we expect that OptiMuS performs well if its composition is organ-optimized according to the differences in each organ (In page 13, Line 268-274).

Minor

The Dil signal is very strong and the image seems to be overexposed (Fig.5c, d). Will it cause errors in the morphological reconstruction?

>> Dil signal was appeared to be overexposed because of the high contrast level we applied with LUT to clearly represent the glomerulus. Indeed, **Dil signal was not saturated** because, during LSFM imaging, the exposure time and other parameters were kept below the saturation level to avoid the saturation of each pixel. We obtained images in 16-bit with an sCMOS camera. Thus, regardless of the “cosmetically” applied contrast level with LUT, the original intensity of each pixel during acquisition was not altered (**Review comment Fig. 1**). We then binarized the image using the Otsu thresholding method. Again, the original intensity value in each pixel is unaltered, thus the result after thresholding is the same regardless of the contrast.

Review comment table 1

Reagent		Refractive index (RI)
OptiMuS		1.47
CUBIC	CUBIC L (Reagent 1)	NA
	CUBIC R (Reagent 2)	1.48 – 1.49
ScaleS	S0	1.375
	S1	1.405
	S2	1.404
	S3	1.418
	S4	1.439
ScaleSQ(0)		1.439
SeeDB2G		1.46
FOCM		1.495
MACS	MACS R0	1.40
	MACS R1	1.48
	MACS R2	1.51

Review comment table 2

200 – 300 µm thick mouse and rat brain	Size	FOCM paper	OptiMuS		
	CLARITY	22 % expansion	No change		
	CUBIC	15.9 % expansion	3.7 % expansion		
	ScaleS	24.5 % expansion	-		
	ScaleSQ	-	5 % expansion		
	SeeDB2	-	12.4 % shrinkage		
	FOCM	2.12 % expansion	15.4 % shrinkage		
	MACS	-	8.5 % shrinkage		
	OptiMuS	-	No change		
1-2 mm thick mouse and rat brain	Size	MACS paper	SeeDB2 Paper	PMID 30155510	OptiMuS
	CLARITY	-	No change	-	1.5 % shrinkage
	CUBIC	No change	10 % shrinkage	10 % shrinkage	17 % expansion
	ScaleS	10 % expansion	3 % shrinkage	15 % expansion	11 % expansion
	ScaleSQ	-	-	30 % expansion	8.5 % expansion
	SeeDB2	8 % shrinkage	No change	-	6.3 % shrinkage
	FOCM	-	-	-	24.5 % shrinkage
	MACS	3 % expansion	-	-	20 % shrinkage
	OptiMuS	-	-	-	No change
Whole mouse brain	Size	MACS paper	FDISCO paper	OptiMuS	
	CUBIC	9 % expansion	15 % expansion	30 % expansion	
	ScaleS	12 % expansion	-	6.8 % expansion	
	SeeDB2	No change	-	-	
	MACS	5 % expansion	-	15.1 % expansion	
	FOCM	-	-	17.6 % shrinkage	
	OptiMuS	-	-	5.9 % expansion	

Review comment table 3

	Transmittance (at 600 nm)	MACS	SeeDB2 paper	FDISCO paper	Other clearing paper (PMID 30155510)	OptiMuS
1-2 mm thick mouse and rat brain	CLARITY	-	-	-	-	64 %
	CUBIC	72 %	-	-	50 %	71 %
	ScaleS	20 %	-	-	25 %	44 %
	ScaleSQ	-	-	-	15 %	41 %
	SeeDB2	4 %	-	-	-	25 %
	MACS	66 %	-	-	-	61 %
	FOCM	-	-	-	-	23 %
	OptiMuS	-	-	-	-	70 %
Whole mouse brain	CUBIC	50 %	-	30 %	35 %	59 %
	ScaleS	3 %	5 %	-	3 %	29 %
	SeeDB2	5 %	12 %	-	-	-
	MACS	50 %	-	-	-	50 %
	FOCM	-	-	-	-	24 %
	OptiMuS	-	-	-	-	56 %

Review comment table 4

Clearing time	MACS paper		OptiMuS	
	Mouse (8-12 weeks)	Rat (8 weeks)	Mouse (8 weeks)	Rat (3 weeks)
Heart	2 d	2 d	0.5 d	1 d
Intestine	-	0.5 d	7 hr	0.5 d
Kidney	2 d	3 d	2 d	-
Liver	3 d	7 d	32 hr	3 d
Lung	3 d	7 d	32 hr	-
Spinal cord	2.5 d	-	0.5 d	-
Spleen	1 d	2 d	0.5 d	1.5 d

Review comment Figure 1

REVIEWERS' COMMENTS:

Reviewer #3 (Remarks to the Author):

For the revised manuscript, the authors give more descriptions and great improvements. The comments are given as below:

1. OptiMus is developed from the combination of iohexol, urea, and D-sorbitol which have already been proved to be effective in the optical clearing. There is no doubt that OptiMus outperforms most previous optical clearing methods to some extent. However, the proposal of this method has limited contribution to the field of optical clearing. The technical integration of OptiMus and OCT is a highlight. There are still some questions that need to be confirmed. Optical coherence tomography (OCT) is an imaging technique that relies on the intrinsic scattering properties of biological tissues to generate imaging contrast. Optical clearing methods could enhance OCT imaging depth. However, the properties of the tissue rather than the size, such as the protein density, will also change during the clearing process, which will have an impact on the OCT imaging results. The statement of this part should be more rigorous. Providing some verification data will be more convincing.

2. For the whole-brain transparency, only the horizontal view is shown in Fig. 4, and only the 900 μm depth signal is given in Supplementary Fig. 7. These are insufficient to prove that OptiMus can realize whole brain transparent imaging.

REVIEWERS' COMMENTS:

Reviewer #3 (Remarks to the Author):

OptiMus is developed from the combination of iohexol, urea, and D-sorbitol which have already been proved to be effective in the optical clearing. There is no doubt that OptiMus outperforms most previous optical clearing methods to some extent. However, the proposal of this method has limited contribution to the field of optical clearing. The technical integration of OptiMus and OCT is a highlight. There are still some questions that need to be confirmed. Optical coherence tomography (OCT) is an imaging technique that relies on the intrinsic scattering properties of biological tissues to generate imaging contrast. Optical clearing methods could enhance OCT imaging depth. However, the properties of the tissue rather than the size, such as the protein density, will also change during the clearing process, which will have an impact on the OCT imaging results. The statement of this part should be more rigorous. Providing some verification data will be more convincing.

>> OptiMus is not an entirely novel clearing method but rather a superior and convenient alternative that overcomes the shortcomings of the existing aqueous-based clearing methods and achieves high transparency of thick tissues and whole organs rapidly while preserving the size and endogenous fluorescent signals. We believe OptiMus is an important contribution to the field of tissue clearing and as the first choice of the general optical clearing process, it certainly helps to visualize 3D volume samples and to be used for 3D pathology analysis of various organs.

The reviewer is concerned about the impact of clearing process on the OCT imaging. OCT uses the interference pattern depending on the phase difference between the reference light and the light that is reflected from the sample. Due to light scattering within biological tissues, however, the quality of OCT images drops significantly with increasing penetration depth. That is to say, the depth of penetration for the light into a biological tissue depends on the scattering characteristics and absorptivity of the tissue. Thus, the OCT imaging depth and contrast are affected by the total spectral-dependent attenuation coefficient of the sample.

To overcome this limitation, the idea that uses tissue clearing reagents such as methyl benzoate, glucose, glycerol, etc. with OCT to reduce scattering and improve the OCT imaging depth has been experimentally investigated by several groups. Our colleague who is collaborating OCT-clearing project with us also showed that OCT with clearing increases the imaging depth, allowing imaging of a much deeper structure (PMID 30814611, See Reviewer Fig. 1 and 2 below). Surely, as the reviewer suggested, too much clearing would result in a decrease in reflectivity thus meticulous optimization is critical to balance these two factors.

Despite the previous efforts, however, due to mal-preservation of tissue size and relatively slow clearing, tissue clearing-based OCT has only limited applications so far. Yet preliminary, we were able to register several thick OCT images of OptiMus-cleared sections together to construct label-free whole 3D organ images. Thus, we believe that the technical integration of OptiMus and OCT would be well suited for the need for fast 3D histopathology, as it can exceptionally preserve the size of the tissue and can reduce scattering coefficient within a few minutes. The reviewer commented that providing some verification data of combining OptiMus and OCT would be convincing. We, however, think it is beyond the scope of this study although it is indeed of our future interest, but certainly requires a significant amount of further study to substantiate the validity.

We have modified the Discussion to reflect the above facts (line 298-304).

2. For the whole-brain transparency, only the horizontal view is shown in Fig. 4, and only the 900 μm depth signal is given in Supplementary Fig. 7. These are insufficient to prove that OptiMus can realize whole brain transparent imaging.

>> In this revision, we additionally included coronally sectioned images as well (Fig 4. g,h). In addition, in Supplementary Fig. 7, we have replaced the previous image with a confocal image of a 2,800 μm thick brain column.

Reviewer Fig. 1 (Fig. 1 from PMID 30814611)

Figure 1. Images of a coronal brain slice (left, control; right, formamide treated) taken by (A) digital camera, (B) optical coherence tomography (OCT), and (C) wide field microscope (hematoxylin and eosin histological staining). In (A), the tick marks represent 2 mm. In (B,C), scale bars represent 2 mm and 1 mm, respectively. (D) Illustration of the overlapped boundaries of brain slices after three different clearing techniques: benzyl alcohol and benzyl benzoate (BABB), Clear^T and Scale. Scale bar represents 2 mm. (E) Histogram of the relative tissue volume changes after each of the four clearing techniques.

Reviewer Fig. 2 (Supplementary Fig. 2 from PMID 30814611)

Supplementary Fig. 2. Enhancement of OCT imaging depth in the cleared brain slice. (A) A series of 3D OCT images after treatment with solutions of increasing formamide concentrations showing that clearing reduces light scattering. The volume of these images is $2.5 \times 2.5 \times 1.7 \text{ mm}^3$. Scale bars represent $500 \mu\text{m}$. (B) OCT signal variations along tissue depth in relation to the degree of tissue clearing. Inset shows an OCT reflectivity image. Plotted data are averaged OCT signals in the cortex ROI delineated by the red rectangle. Scale bar represents 2 mm. A.U., arbitrary units; $I(z)$, light intensity at distance z ; ρ , reflectivity.